# Individual differences in honey bee behavior enabled by plasticity in brain gene regulatory networks

Beryl M Jones[1†]*, Vikyath D Rao[2,3], Tim Gernat[2,4], Tobias Jagla[4], Amy C Cash-Ahmed[2], Benjamin ER Rubin[5], Troy J Comi[5], Shounak Bhogale[6], Syed S Husain[7], Charles Blatti[2], Martin Middendorf[4], Saurabh Sinha[2,6], Sriram Chandrasekaran[7,8]*, Gene E Robinson[1,2,9,10]*

[1]Program in Ecology, Evolution, and Conservation Biology, University of Illinois at Urbana–Champaign, Urbana, United States; [2]Carl R. Woese Institute for Genomic Biology, University of Illinois at Urbana–Champaign, Urbana, United States; [3]Department of Physics, University of Illinois at Urbana–Champaign, Urbana, United States; [4]Swarm Intelligence and Complex Systems Group, Department of Computer Science, Leipzig University, Leipzig, Germany; [5]Lewis-Sigler Institute for Integrative Genomics, Princeton University, Princeton, United States; [6]Center for Biophysics and Quantitative Biology, University of Illinois at Urbana–Champaign, Urbana, United States; [7]Department of Biomedical Engineering, University of Michigan, Ann Arbor, United States; [8]Center for Computational Medicine and Bioinformatics, University of Michigan, Ann Arbor, United States; [9]Neuroscience Program, University of Illinois at Urbana–Champaign, Urbana, United States; [10]Department of Entomology, University of Illinois at Urbana–Champaign, Urbana, United States

*For correspondence:
jonesberylm@gmail.com (BMJ);
csriram@umich.edu (SC);
generobi@illinois.edu (GER)

Present address: †Department of Ecology and Evolutionary Biology, Lewis-Sigler Institute for Integrative Genomics, Princeton University, Princeton, United States

Competing interests: The authors declare that no competing interests exist.

**Abstract** Understanding the regulatory architecture of phenotypic variation is a fundamental goal in biology, but connections between gene regulatory network (GRN) activity and individual differences in behavior are poorly understood. We characterized the molecular basis of behavioral plasticity in queenless honey bee (*Apis mellifera*) colonies, where individuals engage in both reproductive and non-reproductive behaviors. Using high-throughput behavioral tracking, we discovered these colonies contain a continuum of phenotypes, with some individuals specialized for either egg-laying or foraging and 'generalists' that perform both. Brain gene expression and chromatin accessibility profiles were correlated with behavioral variation, with generalists intermediate in behavior and molecular profiles. Models of brain GRNs constructed for individuals revealed that transcription factor (TF) activity was highly predictive of behavior, and behavior-associated regulatory regions had more TF motifs. These results provide new insights into the important role played by brain GRN plasticity in the regulation of behavior, with implications for social evolution.

## Introduction

Understanding the genomic regulatory architecture of phenotypic plasticity is necessary to achieve comprehensive knowledge of the mechanisms and evolution of complex traits. While a growing body of knowledge exists on specific regulatory mechanisms involved in developmental plasticity, less is known about the regulation of behavioral plasticity. Behavioral plasticity is of special interest and presents unique challenges, as behavioral traits derive from the integrated actions of genetic, transcriptomic, and neuronal networks (*Sinha et al., 2020*).

Over the past 20 years, a close relationship between behavioral variation and brain gene expression has been documented across a range of organisms and behaviors (e.g. *Zayed and Robinson, 2012*). Still, the regulatory architecture underlying connections between the genome, brain, environment, and behavior are not well resolved, in part because behavior is itself a complex phenotype with substantial variation between individuals. To fully understand how genomic and transcriptomic variation is transduced into behavioral plasticity, we need both high-dimensional behavioral data at the individual level as well as information on regulatory genomics for those same individuals.

Modification of gene regulatory networks (GRNs) has emerged as an important driver of plasticity during the development and evolution of morphological phenotypes. For example, gains and losses of *cis*-regulatory elements (e.g. binding sites for transcription factors (TFs)) influence species-specific wing melanization patterns in *Heliconius* butterflies and *Drosophila* flies (*Prud'homme et al., 2006*; *Reed et al., 2011*; *Werner et al., 2010*). Pelvic loss in stickleback fish convergently evolved through deletion of a tissue-specific enhancer of the TF *Pitx1* in multiple natural populations (*Chan et al., 2010*). In other cases, similar morphological novelties arose independently through modification of distinct developmental programs, as observed for beak size variation across clades of finches (*Mallarino et al., 2012*). Recruitment of genes involved in developmental plasticity in the evolution of novel phenotypes is thought to be facilitated by the fact that TFs and other regulatory genes often have great temporal flexibility, with extensive variation in expression across developmental time (*Dufour et al., 2020*).

Similar to its role in morphological variation, plasticity in GRNs is theorized to influence behavioral variation, over both organismal and evolutionary time scales (*Sinha et al., 2020*). Brain gene expression is often responsive to environmental stimuli (*Chandrasekaran et al., 2011*; *Cummings et al., 2008*; *Mukherjee et al., 2018*; *Rittschof et al., 2014*; *Whitfield et al., 2003*) and the regulatory activity of many TFs is context-specific with respect to behavioral state (*Chandrasekaran et al., 2011*; *Hamilton et al., 2019*). In addition, modification of hormone signaling and GRNs in peripheral tissues has effects on brain GRNs and resulting behavior (*Ament et al., 2012*). These results demonstrate that GRNs are plastic not only across developmental timescales but also influence real-time behavioral variation. Still, the link between changes in GRNs and behavioral plasticity is weaker than for developmental plasticity (*Sinha et al., 2020*), and to our knowledge, no empirical studies have linked brain GRN plasticity to individual differences in behavior.

Eusocial insects are ideal for studying how GRN activity influences both developmental and behavioral plasticity at the individual scale. Eusociality is characterized by a reproductive division of labor between queen and worker castes, representing a developmentally plastic polyphenism well-studied in many species (e.g. *Holldobler and Wilson, 1990*; *Michener, 1974*; *O'Donnell, 1998*; *Wheeler, 1986*). Queens are specialized for reproductive functions, including mating and egg-laying, and in species with complex eusociality have levels of fecundity orders of magnitude greater than their solitary ancestors. Workers, on the other hand, typically do not perform reproductive behaviors and in many cases are sterile or unable to mate, instead performing many different non-reproductive behaviors in a colony that are essential for colony growth and development. Species with complex eusociality also often show additional within-caste behavioral plasticity, with individuals specializing on specific subsets of tasks based on differences in worker age, morphology, or genetic predisposition. The extensive behavioral plasticity observed in colonies of eusocial species may be linked to ancestral developmental plasticity (*Kapheim et al., 2020*), highlighting the interconnectedness of gene regulation in both developmental and behavioral phenotypes relevant for social behavior (*Sinha et al., 2020*).

We studied the relationship between brain GRN activity and behavior at the individual scale. We focused on a recently discovered, surprising form of behavioral plasticity among worker honey bees. Honey bee workers do not mate, but they possess functional ovaries and can produce viable haploid eggs. Laying workers (LW) are rare in queenright colonies (*Ratnieks, 1993*; *Visscher, 1996*) but frequent in situations of permanent queenlessness, when colonies lose their queen and then fail to rear a replacement queen. In these cases, up to 50% of workers may activate their ovaries (*Sakagami, 1954*) and some of these workers lay eggs, producing thousands of drones prior to colony death (*Page and Erickson, 1988*). Recently, it was discovered that some LWs engage in both reproductive and non-reproductive behaviors (*Naeger et al., 2013*), a level of behavioral plasticity not previously described in honey bee workers. Studying honey bee workers in LW colonies enables investigation of the molecular architecture of behavioral variation typically only seen when

comparing queens and workers, without the confounds of caste-specific developmental and physiological differences.

Recent advances in machine learning and automatic behavioral tracking have enabled the study of individual behavior for thousands of members within social insect colonies (*Crall et al., 2015*; *Gernat et al., 2018*; *Greenwald et al., 2015*; *Mersch et al., 2013*; *Wario et al., 2015*; *Gernat et al., 2020*). We used automatic behavioral tracking, genomics, and the extensive behavioral plasticity present in honey bee colonies with LW to test the hypothesis that individual differences in behavior are associated with changes in the activity of brain GRNs (i.e. changes in the expression of TFs and their target genes). Our results provide key insights into the mechanisms underlying the regulation of individual differences in behavior by brain GRNs.

## Results

### Extensive variation in behavior across laying workers

To define the behavior of individual bees, we deployed a high-resolution, automatic behavior monitoring system on six LW colonies in which each bee (n = 800 per colony) was individually barcoded, similar to *Gernat et al., 2018*. Our extension of this system identifies the location and heading direction of each individual once per second, and uses convolutional neural networks and machine learning to detect behaviors (*Gernat et al., 2020*). For each individual across seven days of tracking (when bees were 15–21 days old), egg-laying events and foraging trips were detected from images of the hive interior and entrance (*Figure 1A*). A total of 115,281 egg-laying events and 96,086 foraging trips were predicted for the six colonies (*Supplementary file 1*).

Colonies exhibited considerable variation in the proportion of bees engaged in egg-laying and/or foraging. With the exception of colony F, more workers were identified as layers than foragers (*Figure 1B*). Across all colonies, a high proportion of bees were observed laying eggs (54% with at least two egg-laying events on at least one day) or foraging (28% with at least two foraging trips on at least one day) during the recording period, while 10.8% of bees performed both egg-laying and foraging on the same day at least once during the seven days of tracking. A small number of these 'generalist' bees (1.3%; 45 individuals) were exceptional in their consistent high performance of both measured behaviors, with a minimum of two egg-laying events and two foraging trips on the same day, across at least three days. Three-day ethograms of an egg-layer, generalist, and forager are shown in *Figure 1C*. Ovary dissection of a subset of individuals revealed that 100% of specialized egg-layers and generalists had active ovaries (ovary scores of 3–5; *Hess, 1942*), compared with only 54% of the specialized foragers (*Figure 1D*; *Supplementary file 2*). Of the foragers with activated ovaries, 13/14 had five or fewer predicted egg-laying events, compared with generalists and layers, which laid an average of 206 eggs (range: 64–774).

The daily and lifetime behavior of each bee was summarized using two behavioral scores: the 'specialist' score, which describes how specialized an individual was on either egg-laying or foraging, and the 'generalist' score, which describes how much an individual performed both egg-laying and foraging. Scores were derived from daily normalized ranks within colonies to allow comparison across days and colonies with differing overall activity levels; bees that performed neither egg-laying nor foraging across the experiment have both specialist and generalist scores of 0. Scores were mapped onto a two-dimensional color space for visualization of behavior over time (*Figure 2A*; *Figure 2—figure supplements 1–2*; *Supplementary file 1*).

### Influence of worker source colony on behavior

To study the influence of source colony (including genetics and development) on behavior, experimental colonies were assembled with workers from different source colonies headed by unrelated queens. A subset of source colonies (4/6) was pre-screened for worker egg-laying in queenless laboratory cages and showed variation in the timing and extent of egg-laying (*Figure 2—figure supplement 3*). Bees in colonies A-C were derived from colonies with naturally mated queens. Queens of *Apis mellifera* mate multiply with up to ~20 males and produce workers with a mix of paternal genotypes (*Adams et al., 1977*; *Estoup et al., 1994*; *Lobo and Kerr, 1993*); workers derived from these colonies were therefore assumed to be a mix of many patrilines. In contrast, experimental colonies D-F were assembled of workers obtained from two different source colonies, each of which was

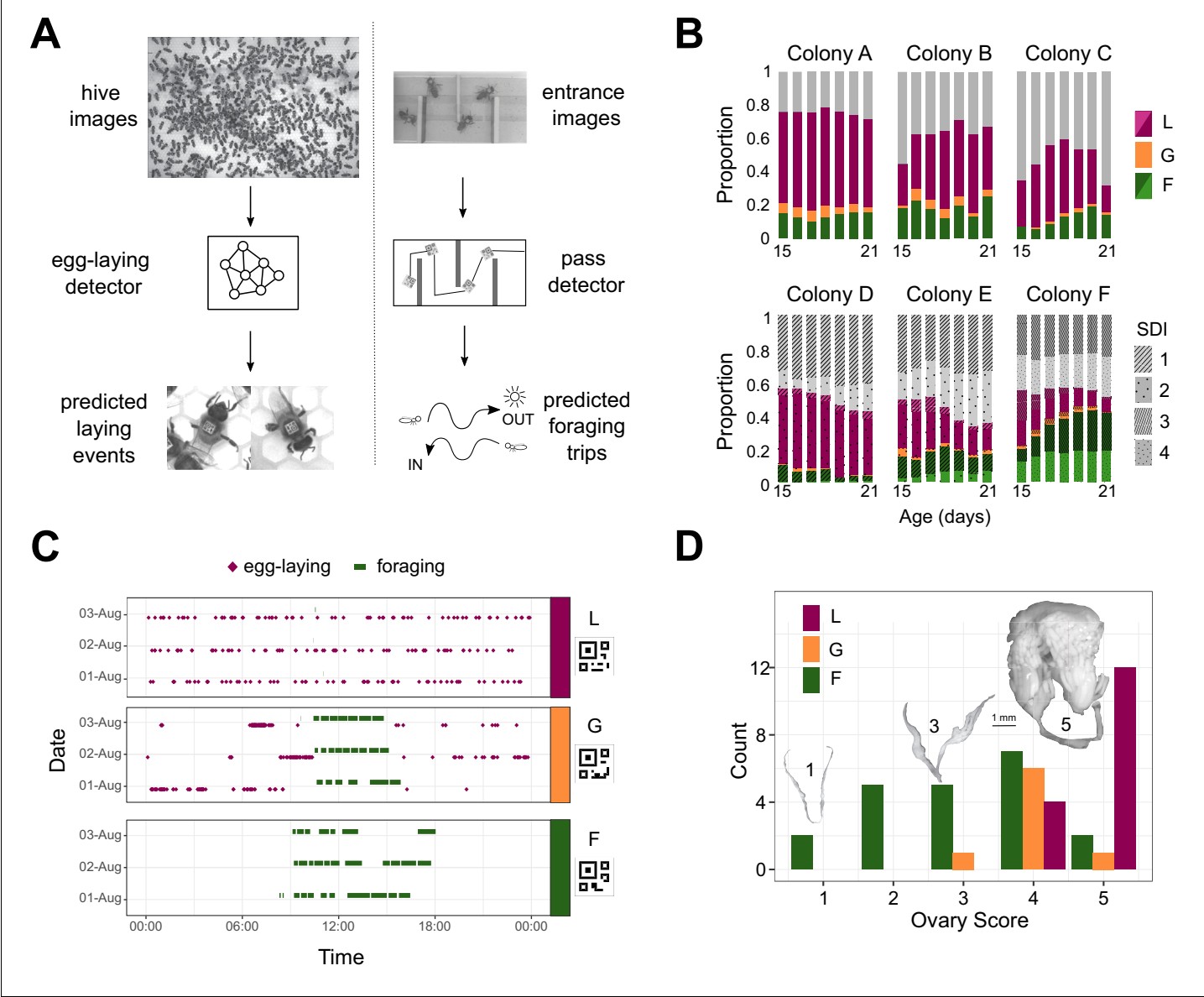

**Figure 1.** Automated monitoring of behavior in queenless colonies of laying worker honey bees. (**A**) Automatic behavior monitoring was performed inside the hive and at the hive entrance to predict egg-laying and foraging events in six colonies (N = 800 bees per colony at the start of each trial). Hive images were captured 1/s for 24 h/day, and entrance images 2/s for 12 h/day beginning when adult bees were 15 days old. (**B**) Proportion of bees alive each day categorized as layers (purple), foragers (green), generalists (orange), or others (gray). For colonies A-C, individuals were from single source colonies headed by a naturally mated queen. For colonies D-F, individuals from two source colonies headed by queens each inseminated by semen from a single different drone (single drone inseminated, SDI) were mixed. Different source colonies are indicated by pattern and hue. (**C**) Ethograms for three individuals selected for sequencing (bCodes shown below group labels) across three days of tracking. (**D**) Distribution of ovary scores for individuals selected for sequencing. Insets are images from bees with ovary scores of 1, 3, and 5. L: layer, G: generalist, F: forager.

headed by a queen artificially inseminated with the semen of a single different drone (SDI). Workers within each SDI source colony are highly genetically related compared with workers from a naturally mated queen colony (average relatedness = 0.75 due to haplodiploidy).

Using SDI colonies allowed us to more easily explore whether the genetic and environmental differences between source colonies would lead to segregation of reproductive and non-reproductive behavior when mixed into the same (queenless) environment. In both colonies D and E, which were replicates of the same two SDI queens' offspring, the behavior of workers differed considerably by source: one source colony (SDI 1) comprised the majority of foragers, while the other (SDI 2)

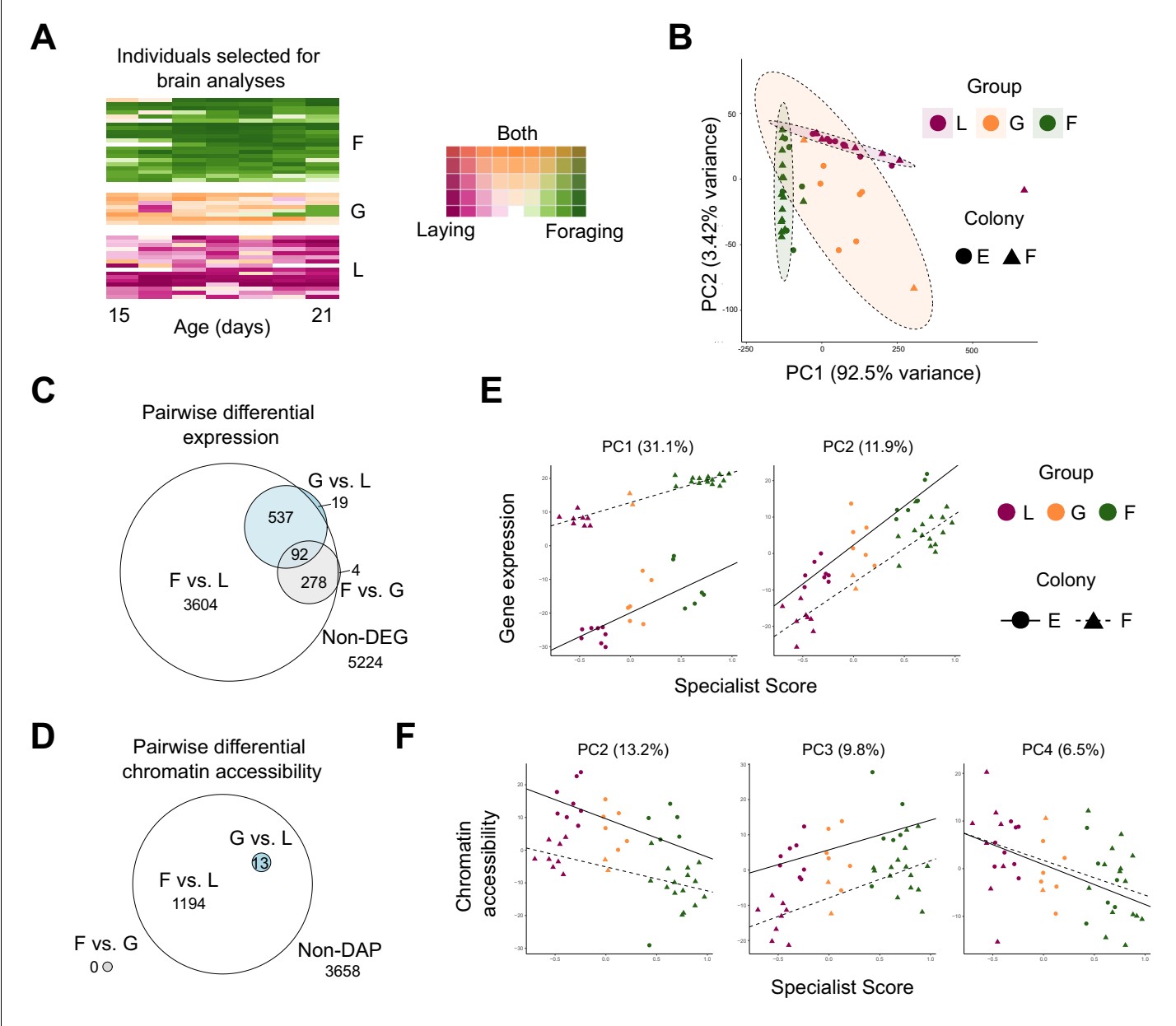

**Figure 2.** Patterns of brain gene expression and chromatin accessibility are associated with behavior. (A) Daily rank-normalized behavior of individuals (rows) selected for brain RNAseq and ATACseq analysis converted to 2D colorspace from specialist and generalist scores. (B) Principal Component Analysis (PCA) of behavioral variation for individuals chosen for brain RNAseq and ATACseq analysis. Metrics included number of eggs laid, number of foraging events, proportion of foraging trips with evidence of nectar collection, proportion of trips with evidence of pollen collection, and proportion of trips with evidence of both nectar and pollen collection. (C) Euler diagram for overlaps of pairwise differentially expressed genes (DEGs) between behavioral groups. Note that one gene was overlapping between F vs. G and G vs. L (but not F vs. L) and is not represented in the diagram due to graphical constraints. (D) Euler diagram for overlaps of genes proximal to pairwise differentially accessible chromatin peaks (DAPs) between behavioral groups. (E) PCs from PCA of brain transcriptomic profiles regressed against specialist score (PC1: $R^2 = 0.947$, $p<0.0001$; PC2: $R^2 = 0.838$, $p<0.001$). (F) PCs from PCA of brain chromatin accessibility regressed against specialist score (PC2: $R^2 = 0.584$, $p<0.001$; PC3: $R^2 = 0.543$, $p<0.0001$; PC4: $R^2 = 0.187$, $p<0.0045$; PC1: $p>0.05$). L: layer, G: generalist, F: forager.

The online version of this article includes the following figure supplement(s) for figure 2:

**Figure supplement 1.** Formulae and color-space mapping for specialist (left) and generalist (right) behavioral scores.
**Figure supplement 2.** Daily behaviors of individual bees (rows) across time in each colony.
**Figure supplement 3.** Smoothed average egg counts for laying workers in laboratory cages.
**Figure supplement 4.** Histogram (bars) and density (lines) of normalized (logCPM) gene expression for genes with (dark gray) and without (light gray) nearby peaks of chromatin accessibility.

contained the majority of egg-layers (*Figure 1B*; *Figure 2—figure supplement 2*). In colony F the two SDI source colony progeny contributed more equally to foraging, while the most specialized egg-laying bees were predominantly from just one source colony (*Figure 1B*; *Figure 2—figure supplement 2*, SDIs 3 and 4). However, even in colonies where SDI source was clearly influential, specialized foragers and layers were identified from both sources, indicating that colony genetics and development are not the only contributors to individual variation in the likelihood of performing these behaviors. Similar patterns of specialization were observed in colonies A-C and D-F (*Figure 1B*; *Figure 2—figure supplement 2*), indicating that they were not an artifact of decreased intracolonial genetic diversity.

## Specialized behavioral groups are highly transcriptionally and epigenetically distinct

A subset of highly specialized foragers, egg-layers, and generalist individuals were selected from two experimental colonies (from only one source SDI colony each, to minimize genetic variation among individuals) for brain gene expression and chromatin accessibility profiling (*Figure 2A*). Sampled individuals were among those with the most extreme specialist and generalist scores within each colony, and were assigned to behavioral groups based upon their lifetime behavior. Principal component analysis (PCA) on behavioral data for these individuals shows these three groups are behaviorally distinct, with generalists intermediate and more variable than forager and layer groups (*Figure 2B*).

Consistent with strong behavioral differentiation, foragers and layers exhibited widespread differences in brain gene expression, with differential expression of nearly half (46%) of all genes expressed in the brain (*Figure 2C*; *Supplementary file 3*). Differences in brain gene expression were much stronger between foragers and layers (4506 differentially expressed genes, DEGs; FDR < 0.05) than for generalists relative to the two specialist groups (648 generalist vs. layer and 374 generalist vs. forager DEGs). Generalists shared transcriptional profiles of both foragers and layers, with nearly all genes differentially expressed between generalists and either specialized group also present on the forager vs. layer DEG list (*Figure 2C*).

Forager vs. layer DEGs were enriched for cytoplasmic translation and transport gene ontology (GO) biological processes, along with many metabolic and biosynthetic processes (*Supplementary file 3*; FDR < 0.05). All enriched GO terms but one (114 of 115) were for genes more highly expressed in foragers relative to layers (forager-biased genes). The only GO term enriched in layer-biased genes relative to foragers, cytoplasmic translation, was also the only enriched GO term for genes overexpressed in generalists relative to foragers. Similarly, GO terms enriched in generalist-biased genes (relative to layers) included many of the transport terms enriched among forager-biased genes (*Supplementary file 3*).

In addition to differences in brain gene expression, layers and foragers showed differences in accessible chromatin in the brain based on the Assay for Transposase-Accessible Chromatin using sequencing (ATAC-seq; *Buenrostro et al., 2013*). 1794 differentially accessible peaks (DAPs; FDR < 0.05) were identified between foragers and layers, proximal to 1207 genes (*Figure 2D*; *Supplementary file 4*). Forager-biased DEGs and genes proximal to forager-biased DAPs overlapped significantly, 1.2x more than expected by chance (p=0.01 for hypergeometric test of overlap). Genes proximal to peaks of accessible chromatin (regardless of differential status) were on average more highly expressed than genes without proximal peaks (p<0.0001, Kolmogorov-Smirnov test), supporting a signal of transcriptional activation near ATAC-seq peaks (*Figure 2—figure supplement 4*). No DAPs were identified for foragers relative to generalists, and there were only 16 DAPs (assigned to 13 genes) for generalists relative to layers (*Figure 2D*). These 13 genes also had DAPs for foragers relative to layers (*Figure 2D*). DAPs between foragers and layers were enriched for 148 GO terms (FDR < 0.05), including developmental processes, morphogenesis, and metabolism (*Supplementary file 4*). Similar to differentially expressed genes, GO enrichment signal came from those DAPs with a bias in foragers (i.e. more accessible in foragers relative to layers); no significantly enriched GO terms were identified from layer-biased peaks, despite 44% of differential peaks being more accessible in layers compared to foragers.

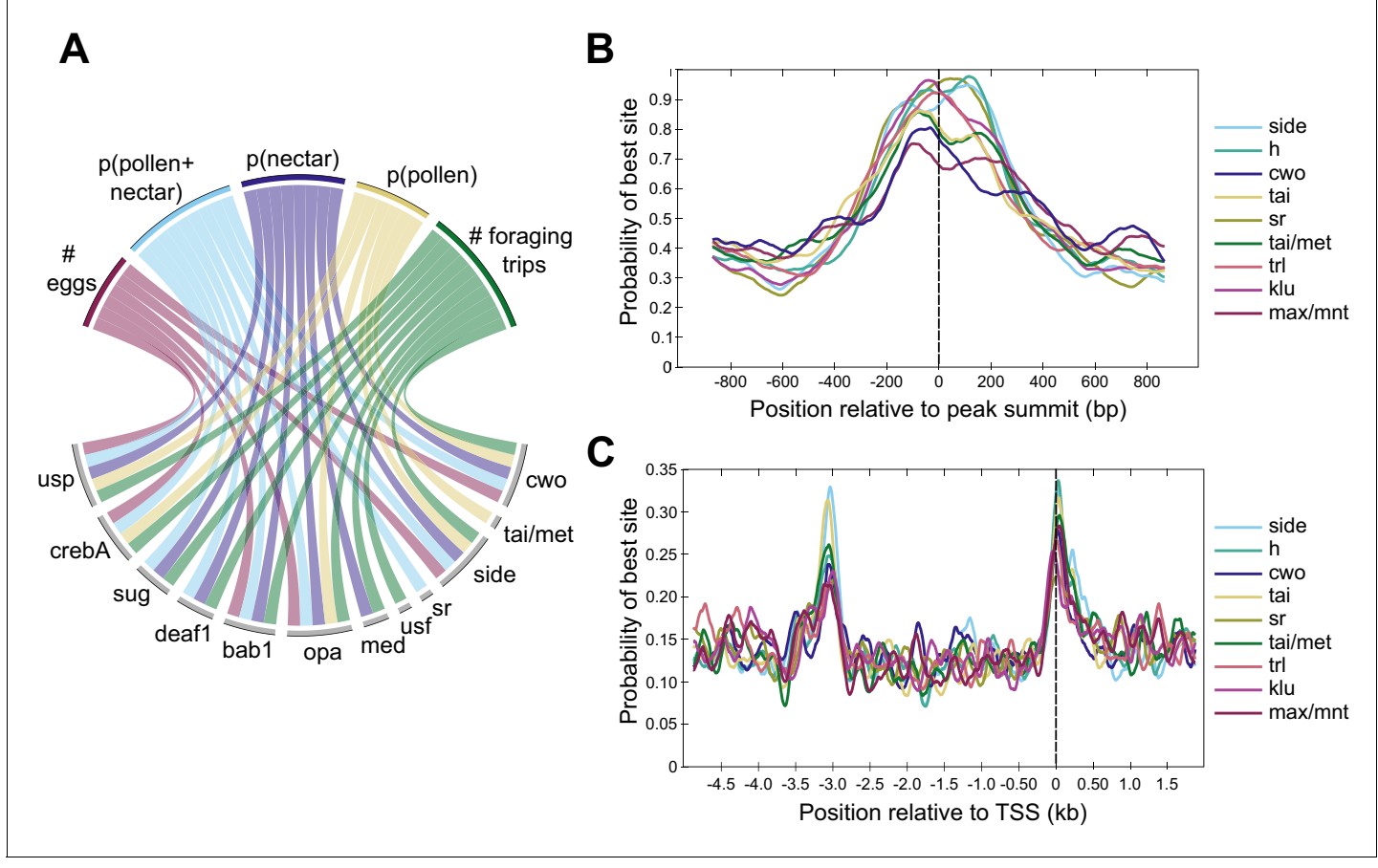

**Figure 3.** Differences in TF activity and TF motif occurrence are associated with specific behavioral phenotypes. (A) Circos plot representing a subset of significant correlations between behaviors (top) and expression of TF modules (bottom). Lines connecting behaviors with TF modules indicate significant associations. TF modules included are those mentioned in the main text or in other figures, and five of nine traits are included for simplicity. All significant correlations between behaviors and TF modules are given in *Supplementary file 8*. For behaviors, p indicates proportion (e.g. p(pollen) is the proportion of returning foraging trips where the bee carried pollen). (B) Motifs enriched within DAPs show maximum binding probabilities near peak summits. (C) Motifs enriched in promoter regions of forager >layer DEGs show elevated binding probabilities ~ 3 kb upstream of and overlapping TSSs. Motif names and sequences are from FlyFactor (*Zhu et al., 2011*) for *Drosophila melanogaster*.

The online version of this article includes the following figure supplement(s) for figure 3:

**Figure supplement 1.** Network of 23 TFs with module expression significantly correlated with nine behavioral and physiological metrics (see *Supplementary file 2*) measured across individuals.

## Brain gene expression and chromatin accessibility are correlated with behavioral variation

Our high-resolution behavioral data allowed us to test whether molecular and behavioral variation were connected not only at the group level, but for individuals as well. Using PCA, we found that degree of individual behavioral specialization was significantly correlated with measures of both brain gene expression and chromatin accessibility (*Figure 2E–F*). Among PCs for gene expression, PCs 1 and 2, which explained 31.1 and 11.9% of the total variance in gene expression, respectively, were significantly correlated with individual behavioral specialist score (*Figure 2E*). Generalists showed intermediate values of these PCs, consistent with an intermediate brain transcriptomic profile. Genes with extreme PC loading values (upper and lower 5% of loadings) for PC1 were enriched for transmembrane and ion transport, functions related to aerobic and cellular respiration, and energy transport (*Supplementary file 5*). PC2 extreme loading genes were enriched for processes relating to detection of light, phototransduction, and sensory perception (*Supplementary file 5*). Extreme loadings for both PC1 and PC2 overlapped significantly with DEGs in the pairwise comparison of layers and foragers (PC1: RF = 1.2, p=1.39e-06, PC2: RF = 1.7, p=8.39e-91).

Similarly, PCA of chromatin accessibility data revealed PCs that were correlated with behavioral variation. Accessibility PCs 2, 3, and 4 were all significantly correlated with the individual behavioral specialist score (*Figure 2F*). Genes with extreme PC loading values for each correlated PC showed enrichment for multiple GO terms, including biological processes related to cell-cell adhesion, locomotion, axon guidance, neuron projection guidance, and synapse organization (*Supplementary file 6*). Many GO terms (11 of 37) were enriched for extreme loading genes of both PCs 2 and 3, while synapse organization was the only enriched term for loadings of PC4.

## GRN activity links molecular and behavioral phenotypes

To test the role of TF and gene regulatory plasticity in the regulation of LW behavioral phenotypes, we conducted TF motif analyses and brain GRN reconstruction (*Chandrasekaran et al., 2011*), individualized for each bee. The activity of many TF modules (TFs and their predicted targets) showed significant correlations with individual variation in several behavioral metrics, including numbers of eggs laid (50 TF modules), number of foraging trips (74 TF modules), and proportion of returning foraging trips with pollen loads (41 TF modules) (*Figure 3A*; *Supplementary file 8*). At the individual level, 23 TF modules were correlated with all nine behavioral and physiological metrics (*Figure 3—figure supplement 1*; *Supplementary file 8*). These behaviorally correlated TF modules include TFs involved in JH signaling (usp, Kr-h1, and Blimp-1), histone acetylation (trx), neuronal remodeling (Kr-h1, Hr51, trx), and circadian rhythms (opa and Hr51).

In addition to the correlations between TF module activity and behavior, many TF motifs were enriched in peaks of differential accessibility or in the regulatory regions of DEGs between specialized layers and foragers. 77 out of 223 motifs (functionally validated in *Drosophila melanogaster*, *Zhu et al., 2011*) were enriched in layer vs. forager DAPs (q-val <0.01, *Supplementary file 7*), and 14 motifs were specifically enriched in the regulatory regions of forager-upregulated DEGs (q-val <0.2, *Supplementary file 7*). Nine motifs were common to both sets (*Figure 3B–C*), including binding sites for TFs involved in regulating nervous system development (hairy, side, sr, and klu), transcription (max/mnt), juvenile hormone (JH) signaling (tai and tai/met), chromatin modification (trl), and circadian rhythms (cwo). These motifs were centrally enriched within DAPs (*Figure 3B*), and showed two peaks of elevated binding probability in the promoter regions of forager-biased DEGs, one ∼ 3 kb upstream of transcription start sites (TSSs) and a second overlapping TSSs (*Figure 3C*). By contrast, only two TF motifs were significantly enriched in the regulatory regions of specialist vs. generalist DEGs (mad and ken, *Supplementary file 7*), and no motifs were enriched within DAPs between generalists and either specialist group (*Supplementary file 7*), likely due in part to the small number of DAPs distinguishing generalists and specialists (*Figure 2D*). Many of the motifs enriched within DAPs or DEG promoters are binding sites for TFs that were themselves differentially expressed, including cwo, tai/met, side, h, and sr (*Supplementary file 3*).

Across individuals, GRN activity was largely consistent within each behavioral group (*Figure 4A*), with TF module activity most distinct between layers and foragers. The relationship between TF expression and behavior was so strong that it was possible to predict individual behavior based solely upon the expression of TFs in the brain using a machine-learning algorithm and leave-one-out cross validation (*Figure 4B*; *Figure 4—figure supplement 1*). TF expression correctly predicted 100% of foragers and 94% of layers. By contrast, it was not possible to predict generalists based on brain TF expression (only 1 of 8 correctly classified).

## Comparative analyses of LW colony behavioral phenotypes and other social insect phenotypes

The performance of both egg laying and foraging by individuals in LW colonies, previously reported in *Naeger et al., 2013*, is unusual for honey bees; these behaviors are otherwise confined to separate castes (queens and workers). This raises the question of whether the mechanisms underlying LW behavior reflect caste-related molecular differences. We compared our gene expression results to previous studies of queens, workers, and worker subcastes in various species of social insects to ask whether the molecular architecture of LW phenotypes may be useful in the context of understanding additional social phenotypes.

In comparison with honey bee subcastes, forager-biased genes in LW colonies showed significant overlap with forager-biased genes in two studies of queenright colonies (when compared with

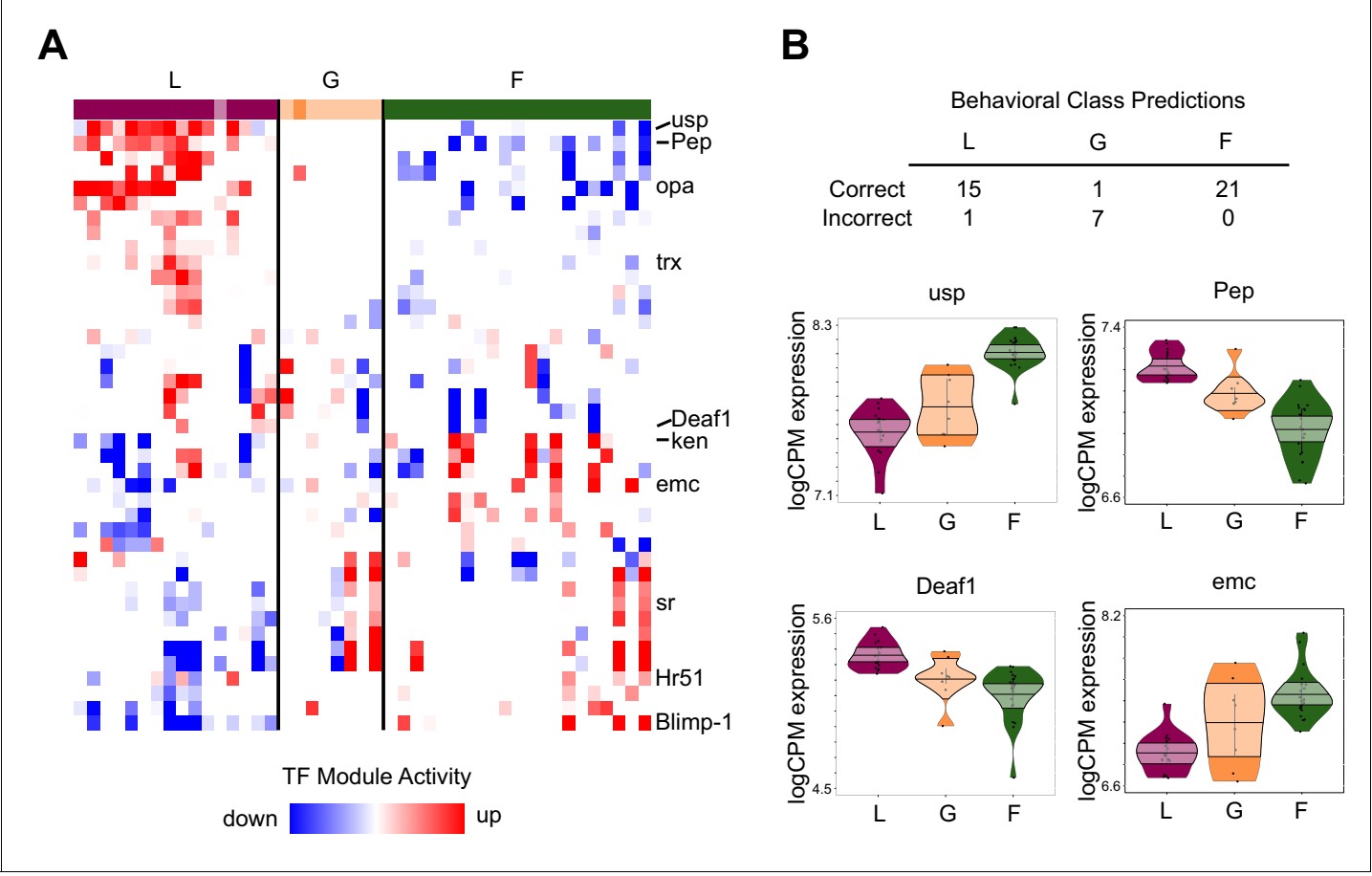

**Figure 4.** TF module activity and TF expression predict individual variation in behavior. (**A**) TF modules (rows) with significant up/downregulation in at least 10 individuals, sorted by hierarchical clustering. Individuals (columns) are ordered by specialist score, with darkly colored blocks indicating correctly classified individuals based on TF expression prediction analysis and lightly colored blocks indicating incorrect classification. TF modules showed patterns of differentiation between L and F, while G were more variable in module activity. Labeled modules are those with TFs shown in panel (**B**) or discussed in text. (**B**) Class prediction analysis based on brain TF expression correctly classified all but one specialist (L: layer, F: forager) but only one generalist (G). Normalized expression (logCPM) of 4 of the top 20 informative TFs for class prediction analysis are shown (others in *Figure 4—figure supplement 1*). Median of points is represented by bold horizontal line within shaded 95% confidence interval, with length of shape and smoothed curve showing range and density of data, respectively.

The online version of this article includes the following figure supplement(s) for figure 4:

**Figure supplement 1.** Normalized expression (logCPM, scaled to a maximum of 1 to allow for comparison across TFs) of the top 20 most informative TFs for class prediction analysis plotted against individual specialist score.

nurses) (RF = 1.7 p=1.707e-09; *Alaux et al., 2009*; RF = 1.9 p=1.740e-07; *Whitfield et al., 2003*; *Supplementary file 9*). Layer-biased genes in this study overlapped with genes upregulated in nurses relative to foragers in these queenright colonies (RF = 1.7, p=3.656e-10; *Alaux et al., 2009*; RF = 2.0, p=1.116e-13; *Whitfield et al., 2003*; *Supplementary file 9*).

In addition, differences in brain gene expression between egg-layers and foragers mirrored caste-related differences across species. Genes differentially expressed between foragers and egg-layers in this study were enriched for previously identified queen vs. worker brain DEGs in *Megalopta genalis* bees, which facultatively engage in both reproductive and non-reproductive behaviors (RF:1.3, p=0.009; *Jones et al., 2017*; *Supplementary file 9*). Overlap was in the expected direction, with queen-biased genes in *M. genalis* overlapping layer-biased genes (RF:2.5, p=0.003) and worker-biased genes overlapping forager-biased genes (RF:1.6, p=0.01). Further, worker-upregulated DEGs in the primitively eusocial wasp, *Polistes metricus* overlapped significantly with forager-upregulated genes in this study (RF:2.6, p<0.0001; *Toth et al., 2010*). In comparison with honey bee

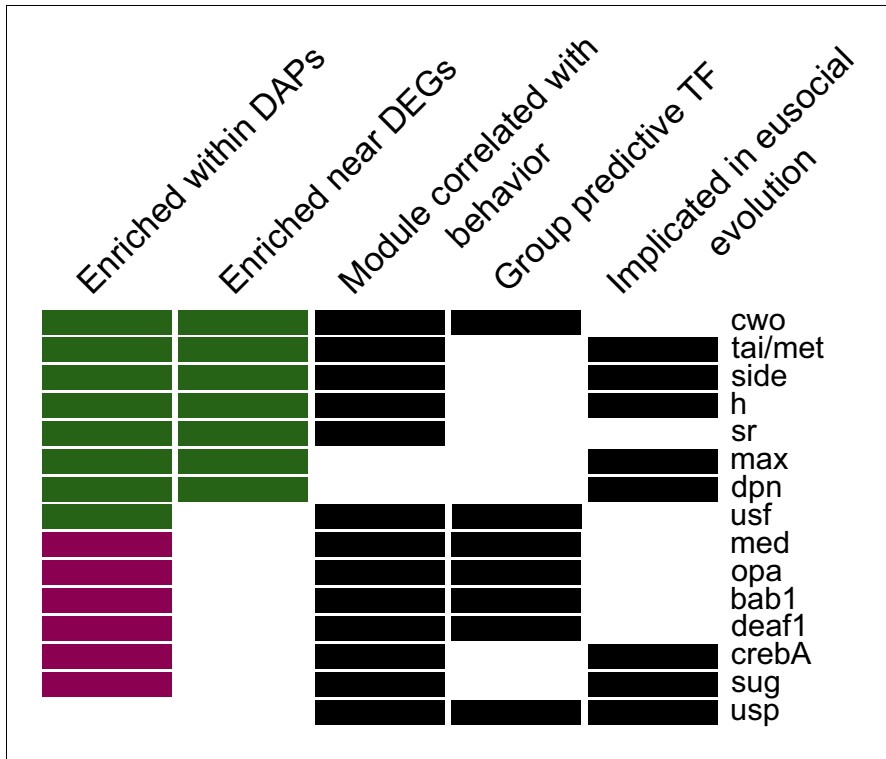

**Figure 5.** Fifteen candidate TFs predicted to regulate egg-laying and foraging behavior based on evidence across all analyses (descriptions of categories in Materials and methods). Names given are for *Drosophila melanogaster* motifs (*Zhu et al., 2011*), with homology to honey bee genes as in *Kapheim et al., 2015*. Color of bar in first two columns indicates whether there was stronger enrichment among forager-biased (green) or layer-biased (purple) peaks or genes.

unmated queens and workers, overlap was significant but in an unexpected direction: queen- and worker-biased genes overlapped with forager- and layer- upregulated genes, respectively (RF:1.4, p=5.495e-08 and RF:1.2, p=0.008; *Grozinger et al., 2007*; *Supplementary file 9*).

Additionally, forager vs. layer DEGs in this study were enriched for genes identified as under selection in two studies of social evolution. Forager vs. layer DEGs overlapped significantly with genes undergoing positive selection in honey bees (RF = 1.1, p=0.015; *Harpur et al., 2014*; *Supplementary file 9*) and across highly eusocial species relative to solitary or primitively eusocial species (*Woodard et al., 2011*; *Supplementary file 9*). Genes under selection in highly eusocial lineages were enriched specifically for genes identified here as upregulated in foragers relative to layers (RF = 1.4, p=0.009), but not for layer-biased DEGs (p=0.106) (*Supplementary file 9*). Forager vs. layer DEGs were not significantly enriched for genes that were identified in a third study as under selection in social lineages of bees (p=0.262; *Kapheim et al., 2015*; *Supplementary file 9*). Many of the forager vs. layer DEGs also found to be undergoing positive selection were related to metabolism (*Supplementary file 9*).

## TFs involved in LW plasticity previously implicated in social evolution

Given that differences in brain gene expression between egg-layers and foragers reflect caste-related differences, we also tested whether there is overlap between TFs involved in LW plasticity and those previously implicated in social evolution. Indeed, many of the TFs we identified above as related to behavioral plasticity based on motif enrichment, group predictive analysis, or brain GRN activity were previously known to be associated with social behavior on an evolutionary timescale. A comparative analysis of the genomes of ten bee species (*Kapheim et al., 2015*) identified 13 TF motifs with associations between binding strength and social complexity. Nine of those 13 motifs were also detected above as enriched within specialist DAPs or DEG regulatory regions (p=0.015, hypergeometric test of overlap), and 6 of those nine are binding sites for TFs included in the above

**Table 1.** Description of 15 candidate TFs regulating specialist behavioral phenotypes in *Figure 5*.

Names given are for *Drosophila melanogaster* motifs (*Zhu et al., 2011*), with homology to honey bee genes as in *Kapheim et al., 2015*. Function summaries are adapted from *D. melanogaster* gene annotations from FlyBase (release FB2020_05; *FlyBase Consortium et al., 2019*). Note that terms related to 'regulation of transcription' apply to most TFs but were omitted for brevity.

| Motif | TF name | Function(s) |
|---|---|---|
| cwo | *clockwork orange* | circadian regulation of gene expression; dendrite morphogenesis |
| tai/met | *taiman, Mondo* | ecdysone receptor co-activator; lipid and carbohydrate metabolism |
| side | *sidestep, E(spl)mgamma-HLH* | pattern specification; neurogenesis; neuronal stem cell maintenance |
| h | *hairy* | cell morphogenesis; tracheal system development; cellular metabolism |
| sr | *stripe* | central nervous system development |
| max | *Max* | cell and organismal growth |
| dpn | *deadpan* | adult locomotory behavior; neuroblast development |
| usf | *Usf* | [unknown] |
| med | *Medea* | dorsal-ventral patterning; activin receptor signaling; eye morphogenesis; germ-line stem cell division and maintenance; neuron development |
| opa | *odd paired* | embryogenesis; midgut development; adult head morphogenesis; neural stem cell development; circadian rhythm |
| bab1 | *bric a brac 1* | pattern formation; ovary morphogenesis; abdominal pigmentation; olfactory receptor neuron fate diversity |
| deaf1 | *Deformed epidermal autoregulatory factor-1* | embryo development; regulation of immune response |
| crebA | *Cyclic-AMP response element binding protein A* | salivary gland development; cuticle development |
| sug | *sugarbabe* | regulates expression of insulin-like peptides and genes involved in lipid and carbohydrate metabolism |
| usp | *ultraspiracle* | cell migration; response to ecdysone; germ cell development; metamorphosis; mushroom body development; neuron remodeling |

individualized GRNs. Along with TF module correlation and behaviorally-predictive TF expression, these results highlight a set of 15 TFs as compelling candidates in social plasticity and evolution, with significant associations in at least 3 of the five analyses (*Figure 5*; *Table 1*). The 15 TFs have functions related to known mechanisms associated with social behavior, including brain development (*Hamilton et al., 2016*), JH signaling (*Woodard et al., 2011*), and chromatin changes via histone acetylation (*Simola et al., 2016*).

## Discussion

Uncovering the regulatory mechanisms involved in behavioral plasticity is important to fully understand how behavioral phenotypes develop and evolve. We used automatic behavioral tracking and genomics to uncover the role of brain GRN activity in the extensive behavioral variation observed in colonies of laying worker honey bees. We discovered that continuous phenotypic variation is associated with continuous variation in both brain gene expression and brain chromatin accessibility, and that TF activity is predictive of behavioral phenotype at the individual level. These results provide

new mechanistic insights into the important role played by brain GRNs in the regulation of behavioral variation, with implications for understanding the mechanisms and evolution of complex traits.

Our high-dimensional behavioral data revealed a near continuous distribution of phenotypes along an axis of egg-laying and foraging, two behaviors that are typically expressed separately in the queen and worker castes of honey bee colonies. Consistent with previous reports of ovary activation in queenless colonies (*Page and Erickson, 1988*; *Sakagami, 1954*), over half of workers tracked laid eggs. Some of these workers also engaged in foraging, consistent with the observations of *Naeger et al., 2013*, which supports the suggestion that some laying workers are not 'selfish' reproducers but engage in activities that may benefit the colony as a whole. We also showed a decoupling between ovary status and behavior for some individuals, unlike what has been observed in many other social insect species (*Barth et al., 1975*; *Michener, 1974*; *Wilson, 1971*). Two-thirds (14/21) of the foragers had activated ovaries, but most laid eggs infrequently or not at all, demonstrating that ovary activation alone is not a strong predictor of exactly which individuals will lay eggs. This decoupling of reproductive physiology from reproductive behavior is consistent with the evolutionary co-option of reproductive signaling pathways for non-reproductive behaviors, a phenomenon well documented in honey bees (*Tsuruda et al., 2008*; *Graham et al., 2011*; *Page et al., 2012*). Given previous demonstrations of cross-talk between peripheral tissues and brain gene networks in the honey bee (*Ament et al., 2012*; *Wheeler et al., 2013*), our results further suggest that behavioral variation in queenless workers likely involves the coordinated actions of multiple tissue types, including the ovary.

Like the task specialization observed in typical, queenright colonies of honey bees and many other social insects (*Oster and Wilson, 1979*), the majority of individuals in LW colonies showed consistency in performance of either egg-laying or foraging, but not both. It is important to note that genetic variation may contribute to individual differences in behavior (*Page and Robinson, 1991*; *Page and Robinson, 1994*). However, the induction of egg-laying behavior in queenless colonies is itself a plastic response, suggesting that at least for egg-laying and generalist individuals, a combination of hereditary and environmental factors likely influence the development of these behavioral phenotypes. Task specialization can contribute to increased efficiency in social insects, either through learning or reduction of task switching costs (*Holldobler and Wilson, 1990*; *Jeanson et al., 2008*; *Trumbo and Robinson, 1997*; c.f. *Dornhaus, 2008*). In queenless colonies of honey bees, specialization along a reproductive/non-reproductive axis may lead to increased production of haploid males prior to the death of workers, with specialized foragers collecting food for these developing drones while specialized egg-layers work to produce thousands of drones synchronously in these terminal colonies (*Page and Erickson, 1988*). These findings suggest that LW honey bees may display a form of colony organization that is adaptive, as opposed to one of chaos and competition, which has long been thought to characterize LW colonies (*Morse, 1990*; *Ratnieks et al., 2006*; *Ratnieks and Wenseleers, 2008*; *Dadant & Sons, 1975*; *Wenseleers and Ratnieks, 2006*). Worker derived drones have viable sperm (*Gençer and Kahya, 2011*) and therefore may provide a permanently queenless honey bee colony with a final fitness opportunity if the males can successfully mate with queens. It is difficult to evaluate this hypothesis because the incidence of permanently queenless colonies is not known in natural populations of honey bees. However, production of drones by workers in LW colonies is similar to that observed in bumble bees, where worker competition over male production is a normal part of the colony cycle after queen death (*Cnaani et al., 2002*; *Free, 1955*), or even prior to queen death in some species (*Velthuis and Duchateau, 2011*).

Consistent with many other studies of behavior and brain gene expression across animal species (e.g. *Bukhari et al., 2019*; *Mello et al., 1992*; *Whitfield et al., 2003*), we identified robust brain transcriptional signatures associated with specific behavioral states. Beyond these group level differences, we also discovered that large components of this molecular variation were correlated with behavior, and both behavior and brain gene regulatory activity were continuous across bees. Our finding that both brain gene expression and chromatin accessibility vary continuously with behavioral phenotype suggests that behavioral plasticity is subserved by continuously varying molecular programs, rather than threshold-based or quantized changes.

At the individual bee level, changes in the expression of TFs, accessibility of TF motifs in enhancers and promoters, and activity of TF module target genes were all strongly associated with behavioral state. This is highlighted by the results of our predictive analysis, where 97% of specialists were accurately predicted to phenotype based on TF expression alone, despite the small number of

TFs relative to all differentially expressed genes. Spatial and temporal integration of discrete events such as TF binding, aggregated at the whole brain level and across TFs and genes, may lead to the continuous variation we observed in gene expression and chromatin accessibility (e.g. *Araya et al., 2014*).

In addition to predicting the collective behavioral phenotypes of individual bees, our analysis of GRNs allowed us to probe the influence of TF module activity on single behaviors. We identified a set of 23 TF modules that were associated with all aspects of behavior and physiology we measured. These TFs appear to coordinate sets of behaviors that are not overtly linked (e.g. proportion of nectar foraging trips and number of eggs laid) but may be influenced by the same regulatory machinery. Three of these modules are activated by TFs downstream of JH, a hormone with numerous well-studied roles in social insect behavior, including the regulation of oogenesis in queens and age-related division of labor in workers (*Tsuruda et al., 2008*; *Hamilton et al., 2016*; *Page et al., 2012*). Our results are consistent with a role of JH signaling in queenless colonies of worker honey bees, regulating a behavioral division of labor between specialized egg-layers and foragers. These findings match previous work describing differences in JH titers between egg-laying workers and foragers in queenless colonies (*Robinson et al., 1992*), and suggest that mechanisms underlying variation in egg-laying behavior may be similar to nurse/forager differences in queenright colonies. Overlap in brain gene expression profiles between nurses and egg-layers further supports this conclusion.

By combining our analysis of GRNs in individual bees with motif enrichment in gene regulatory regions across individuals, we identified a set of 15 TFs which appear to play a key role in regulating specialist behavioral phenotypes (*Figure 5*). Intriguingly, many of these TFs were also identified as relevant for social evolution, with increases in TF motif presence in gene promoters of social compared with solitary species of bees (*Kapheim et al., 2015*). We observed especially strong overlap of these evolutionarily-implicated TFs and those with motif enrichment within differentially expressed genes or differentially accessible chromatin of specialist phenotypes. This suggests that regulatory regions that arise during evolutionary transitions to eusociality may be maintained and even further refined for the regulation of specialized subcastes in social species. In contrast, comparatively little overlap was seen when comparing evolutionarily-implicated TFs with TFs whose expression was most predictive of specialist behavioral phenotypes. This mismatch between TF expression and motif presence may reflect the complexity of GRNs, where genetic and epigenetic landscapes modulate the effects of TF activity. Alternatively, these results may reflect differences in the mechanisms underlying intra- vs. interspecific variation in social behavior. Further research exploring the role of these TFs and their activity in a range of contexts is needed to provide clarity on these results.

While behavioral specialization appears to be common among members of queenless honey bee colonies, the finding of even a small number of generalist bees who perform both egg-laying and foraging has intriguing implications. The presence of these generalists suggests that despite the long divergence from a solitary ancestor (~85 my, *Branstetter et al., 2017*), honey bees retain great flexibility for performance of multiple behaviors that are typically confined to either the queen or worker caste. Latent plasticity in social insects that is inducible under extreme conditions is also seen in morphologically and temporally defined worker subcastes under queenright conditions (*Robinson, 1992*; *Simola et al., 2016*; *Wilson, 1980*). Generalists showed high variation in behavior, and similarly were difficult to predict phenotypically based on TF activity, unlike specialists. Further, brain GRN activity in these individuals was less defined, with fewer TF modules showing significant up- or down-regulation in generalist individuals compared with specialists. Combined with PCA on brain gene expression and chromatin accessibility, these findings suggest that generalists are molecularly intermediate between specialized groups.

Our discovery of intermediate generalist phenotypes in laying worker colonies, along with their molecular signatures, provides support for one of the leading theories of eusocial evolution, the Ovarian Ground Plan Hypothesis (OGPH). The OGPH posits that the emergence of queen and worker castes from solitary ancestors involved the genetic decoupling of reproductive and non-reproductive behavioral programs through changes in gene regulation acting on ancestral plasticity (*Gadagkar, 1997*; *Turillazzi and West-Eberhard, 1996*; *West-Eberhard, 1987*). The phenotypic continuum we observed in laying worker colonies, with both reproductive and non-reproductive specialists as well as generalists, suggests that this decoupling process is at least partially reversible and/or incomplete in honey bees, unlike in eusocial species where workers lack reproductive anatomy and corresponding behaviors entirely (e.g. ants and higher termites). Additionally, molecular

characterization of this behavioral variation, especially our TF analyses, supports the hypothesis that incremental changes in gene regulatory network activity led to the decoupling of solitary behavioral programs into distinct queen and worker castes. This hypothesis is consistent with previous research linking changes in TF activity with social evolution (*Kapheim et al., 2015*; *Kapheim et al., 2020*). If correct, this hypothesis provides a framework for understanding the evolution of eusociality at the molecular level.

# Materials and methods

**Key resources table**

| Reagent type (species) or resource | Designation | Source or reference | Identifiers | Additional information |
|---|---|---|---|---|
| Biological sample (*A. mellifera*) | queens | Honey Bee Insemination Service, Washington State University | | SDI colonies |
| Biological sample (*A. mellifera*) | workers | University of Illinois Bee Research Facility | | |
| Commercial assay or kit | RNeasy Mini Kit | Qiagen | 74104 | |
| Commercial assay or kit | TruSeq Stranded mRNA HT kit | Illumina | RS-122–2103 | |
| Commercial assay or kit | Nextera DNA Sample Preparation Kit | Illumina | FC-121–1031 | |
| Software, algorithm | Trimmomatic | | RRID:SCR_011848 | v0.36 |
| Software, algorithm | STAR | | RRID:SCR_015899 | v2.5.3 |
| Software, algorithm | bwa | | RRID:SCR_010910 | v0.7.17 |
| Software, algorithm | Picard | | RRID:SCR_006525 | v2.10.1 |
| Software, algorithm | R project for statistical computing | R Core Team | RRID:SCR_001905 | |

## Bees and colony setup

### Source colonies

Honey bee colonies were maintained according to standard beekeeping practices at the University of Illinois Bee Research Facility in Urbana, Illinois. One-day-old adult worker bees were obtained by removing sealed honeycomb frames of late-stage pupae from source colonies in the field and housing them in an incubator inside emergence cages at 34°C and 50% relative humidity. Bees were removed from frames daily to collect adults less than 24 hr old.

Prior to establishing the colonies of barcoded bees, 16 source colonies were screened for worker egg-laying ('laying worker', LW) potential by stocking Plexiglas cages with 50–100 one-day-old workers and holding them in queenless, broodless conditions. Cages contained small pieces of 3D-printed honeycomb (similar to *Fine et al., 2018*) to provide a standardized location for workers to lay eggs, as well as 50% sucrose solution and pollen paste (45:45:10 ratio by weight of pollen, honey, and water) provided *ad libitum* and refreshed daily. Cages were monitored daily to count eggs. We found, as in other studies, variation in the timing and extent of LW development among different source colonies (*Figure 2—figure supplement 3*), reflecting the effect of genotypic and/or environmental differences on laying worker potential (*Miller III and Ratnieks, 2001*; *Page and Robinson, 1994*; *Robinson et al., 1990*; *Velthuis, 1970*). When possible, source colonies were chosen from among those screened that displayed high levels of worker egg-laying in cages within 14 days.

To reduce genetic variation among bees used for sequencing, experimental colonies D-F were established from a mix of two source colonies each headed by a queen of either *A. mellifera ligustica* or *A. mellifera carnica* origin who had been artificially inseminated with semen from a single drone (SDI) (queen rearing and inseminations performed by Sue Cobey, Honey Bee Insemination Service; Washington State University; US stocks of bees are primarily, but not completely *ligustica* or *carnica*).

Experimental colonies A-C were established from naturally mated, *A. mellifera ligustica* source colonies. Honeycomb frames of late-stage pupae were removed from source colonies and maintained in an indoor incubator. Worker bees were collected from these frames each day to obtain 0–24 hr old individuals for barcoding. A total of 800 bees were used for each experimental colony, collected and barcoded over 1–2 days upon eclosion (*Supplementary file 10*).

### Barcoding bees

Bees were tagged with 'bCode' barcodes as in *Gernat et al., 2018*. Unique sets of bCodes were used to differentiate bees barcoded on different days, as well as to differentiate bees from different source colonies in colonies D-F. To attach bCodes to bees, workers were anesthetized on ice and then positioned using soft forceps (BioQuip, Compton, CA). A small drop of Loctite Super Glue Gel Control (Henkel, Düsseldorf, Germany) was applied to the center of the thorax of each bee, followed by a bCode positioned with its left and right edge parallel to the anteroposterior axis of the bee. Bees were carefully placed in plastic dishes until they recovered from cold anesthetization, at which point the glue was dry. After waking, all bees were placed in a large container with Fluon-coated walls (Insect-a-Slip, BioQuip) where honey was provided ad libitum until placement into a custom observation hive, described below. At the end of each barcoding day, bees were carefully transferred into the observation hive.

## Behavioral tracking

### Hive monitoring

Barcoded bees were housed in a glass-walled observation hive with a one-sided plastic honeycomb frame, as in *Gernat et al., 2018*. Bees were unable to access the back side of the honeycomb, and could exit the hive through a plastic tube to the outside. Colonies were maintained in a dark room with a heater and humidifier that kept the room at approximately 32°C and 50% relative humidity.

Infrared light (not visible to bees) was used to illuminate the hive from both the front and back while capturing hive images. Images were acquired at one-second resolution with a monochrome Prosilica GX6600 machine vision camera (Allied Vision, Stadtroda, Germany) fitted with a Nikkor AF 135 mm f/2 D DC prime lens (Nikon, Minato City, Japan). Additional details about image acquisition can be found in *Gernat et al., 2018*. Images were saved to a redundant array of independent disks, then copied onto a computing cluster (Biocluster, UIUC) for analysis after the end of each experimental recording period.

### Entrance monitoring

Colonies of barcoded bees were given access to the outside via a tube connected through an exterior wall of the Bee Research Facility to an entrance equipped with an automated flight activity monitor as in *Geffre et al., 2020*. This monitor included a maze to slow down incoming and outgoing bees, and a Raspberry Pi camera (five megapixel v1.3, Adafruit, New York, NY) that imaged the maze twice per second from 07:00 until 19:00 daily. The camera was controlled by a Raspberry Pi 2B computer running the Raspian eight operating system. Images were acquired using the raspistill program and the following options: -n -ISO 400 w 2593 h 1400 -cfx 128:128 -x none -e jpg -q 90 -tl 500 t 595000 -bm.

## Barcode detection

Barcodes were detected in hive images as in *Gernat et al., 2018* and filtered to facilitate subsequent behavioral analyses. Filtering involved removal of potential tracking errors, including removal of barcodes that did not pass read error correction. In addition, records for barcodes that were read twice in the same image were removed, as were hive image records of the same barcode identified more than 5 cm/second between successive detections, which are likely to be misidentifications. An average of 94.51% of detections remained after these filtering steps (range across colonies: 91.94–97.11%). Finally, the time of death of each bee was estimated using the last time she was observed for at least 4 min during a 5 min window above the third row of honeycomb cells from the bottom of the hive; dead bees tend to accumulate below this level prior to being removed by other bees (*Gernat et al., 2018*). Records for bees following their time of death were filtered out so behavioral scores (below) were calculated only over times in which bees were alive.

In entrance monitor images, barcodes were similarly detected as in hive images, but with parameters adjusted for images produced by the Raspberry Pi camera. Fast-moving bees were not filtered out in entrance images, because bees do move quickly through the entrance monitor and due to the relatively small number of bees that fit into the maze, spurious fast movement due to bCode decoding errors is unlikely.

## Egg-laying detector
### Annotated image library

Hive images from three experimental colonies and across 12 different days were used for manual annotation of egg-laying events. The software Fiji (*Schindelin et al., 2012*) was used to mark the bCode positions of all workers laying eggs in an initial set of 1500 hive images, followed by an additional set of 782 images, each annotated by three independent observers. After the initial identification of egg-laying bees in these images, the two seconds before and after each egg-laying event were also annotated for those bees. Bees not marked as laying eggs with visible bCodes were considered non-egg-laying for training of the CNN, below.

### CNN training and performance estimation

Two convolutional neural networks (CNNs) were trained on the annotated egg-laying images, using TensorFlow (*Abadi et al., 2016*). Methods are described fully in *Gernat et al., 2020* and are presented briefly here. The first CNN used images cropped to include just a small rectangular region behind the barcode of each bee. For egg-laying bees, these images show the honeycomb, because their abdomen is backed into the comb and thus not visible. For non-layers, these images show the abdomen. The CNN was trained to differentiate between these two cases. The second CNN was applied to images of bees that were identified as potential egg-layers by the first CNN. It used slightly larger images that showed the entire bee and was trained to use information about the bee's posture and her immediate surroundings to identify false positives, which were subsequently filtered out.

Application of a CNN to an image results in a score between 0 and 1 that reflects the likelihood of that image showing the event of interest. Deciding whether a score is sufficiently high for assuming that the event took place involves thresholding that score. To choose thresholds for each CNN score and a minimum egg-laying duration, a calibration set of images, which were not used for training the CNNs, was used to estimate the performance of the egg-laying detector for different threshold combinations. Thresholds were chosen from this calibration set to maximize the detector's positive predictive value, then were applied to an independent test set of images that had also never been seen by the detector to obtain unbiased performance values. Based on the performance estimation on the test set of images, the egg-laying detector had the following performance: 99.71% accuracy, 35.39% sensitivity, 100% specificity, 100% positive predictive value, and 99.71% negative predictive value. Minimizing false positives came at a cost to sensitivity, but bees who lay eggs will likely do so more than once over the course of the experiment and can thus still be identified as egg-layers (honey bees possess multiple ovarioles, each of which can develop eggs simultaneously [*Hess, 1942*]). Egg-laying detections were further aggregated into events: subsequent detections that occurred within 10 s and 11.2 mm (the width of two honeycomb cells) of one another were assumed to belong to the same egg-laying event and were merged.

## Filtering and annotation of entrance data

Raw detections of bees in the entrance were filtered as in *Geffre et al., 2020*. Briefly, a bee must traverse at least one-third the distance of the entrance monitor to be counted, and traversals that occurred within 10 s of each other were merged into a single event. These traversal events were then determined to be incoming or outgoing based on the positional coordinates of the bee at the start and end times of each event. Numbers of foraging trips (*Supplementary file 1*) was inferred from series of outgoing and incoming events.

Incoming foraging trips were additionally annotated with trophallaxis data to determine whether a forager likely returned with nectar. CNNs trained to identify pairs of bees engaged in trophallaxis as well as the direction of trophallaxis (i.e. which bee was donor and which was recipient; *Gernat et al., 2020*) were used to annotate incoming trips for all bees. Parameters used for the

detector resulted in the following performance metrics based on test images: 88.7% sensitivity, 99.6% specificity, 90.4% positive predictive value, 99.6% negative predictive value, and 88.9% accuracy in determining trophallactic role (donor or receiver) of each bee. If a bee was a trophallaxis donor within 5 min after returning from a trip (*Seeley, 2009*), with no trophallaxis reception prior to the donation, that foraging trip was annotated as a nectar trip. Additionally, incoming trips were manually annotated for pollen on the hindlegs of returning bees for colonies D-F. Combining these nectar and pollen data for each trip, the proportion of foraging trips with nectar ('p.nectar'), pollen ('p.pollen') or both ('p.both') were calculated per bee in these colonies.

## Specialist and generalist scores

In order to characterize the activity of egg-laying and foraging for each bee, two behavioral scores were created. The 'specialist' score describes how specialized an individual was on either egg-laying (scores near −1) or foraging (scores near +1) relative to other bees in the colony; bees that consistently performed both egg-laying and foraging, or that performed neither behavior, have specialist scores near 0. The generalist score ranges from 0 to 1 and describes the degree to which an individual performed both egg-laying and foraging behaviors, differentiating bees with specialist scores near 0 based on the performance (or not) of egg-laying and foraging. Scores were created by first counting the number of egg-laying and foraging events per day. Bees were then ranked for each behavior relative to other bees in the colony on the same day, with tying ranks being assigned the minimal rank (e.g. if three bees were tied between the 4th and 8th ranked bees, they all received a rank of 5). Ranks were then normalized by dividing by the maximum rank, so that all ranks were in the range [0,1]. The normalized rank space for each bee (i.e. normalized egg-laying rank and normalized foraging rank) was then mapped to behavioral scores (and corresponding color space) using the following formulae in polar coordinates $(\rho, \theta)$ on the two-dimensional rank space: generalist score = $(1/2)\rho^2 \sin^4 2\theta$, specialist score = $\sin(\theta - \pi/4)\rho^4 \cos^4 2\theta$. Note that the numerical value of the scores has no biological meaning, but is simply a mapping from rank space to the space of colors as shown in *Figure 2—figure supplement 1*.

## Selection of bees for sequencing

The median of specialist and generalist scores was weighted to emphasize the latter part of the experiment; days 15–21 received a weight of 1–7, respectively, and each day's score was multiplied by this weight. These scores were used to characterize the overall behavior of each bee in the colony. The rank approach allowed for normalization across days with different overall levels of activity in the colony, and the median score across days provides an overall assessment of the lifetime behavior of each bee. These weighted median scores were used to rank all bees, and the top ranking specialists and generalists from two colonies were selected for brain RNA sequencing (RNAseq) and Assay for Transposase-Accessible Chromatin using sequencing (ATACseq). Scores for each sequenced bee (n = 45, 25 from colony E, 20 from colony F), as well as total numbers of detected egg-laying and foraging events per bee, are provided in *Supplementary file 2*.

To examine variation in behavior within and among groups, principal component analysis (PCA) was performed on the following set of behavioral traits (see also *Supplementary file 2*): number of eggs laid, number of foraging events, proportion of trips with evidence of nectar collection, proportion of trips with evidence of pollen collection, and proportion of trips with evidence of both nectar and pollen. PCA was performed in R using the prcomp function and plotted using the ggplot2 package.

## Tissue dissection and homogenization

At the end of behavioral tracking, bees were collected from each colony and stored at −80℃. All colonies were collected between 21:00-23:00 to ensure foragers were inside the hive. For bees selected for sequencing, abdomens of each bee were carefully removed on dry ice and incubated for 16 hr at −20℃ in RNA-later ICE (Life Technologies, Carlsbad, CA). Ovaries were imaged and assessed for ovary development using a 1–5 scale adapted from *Hess, 1942* to assign an ovary score; a score of 3–5 indicates ovary activation. These dissections confirmed that egg-layers and generalists had activated ovaries, while many foragers did not. Ovary scores, as well as number of ovarioles as determined from dissections, are given in *Supplementary file 2*.

The head of each bee was freeze-dried at 300 milliTorr for 55 min, and whole brains were removed from the head capsule in a dry ice ethanol bath (*Schulz and Robinson, 1999*). Dissected brains were stored individually in 1.5 mL microcentrifuge tubes at −80°C until extractions.

Brains were individually homogenized in 150 µL phosphate buffered saline (1X PBS, Corning, Corning, NY, cat. #21–040-CV) with protein inhibitor complex (PIC, Complete Tablets, EDTA-free Protease Inhibitor Cocktail from Roche, Basel, Switzerland, cat. #04693132001) using a motorized pestle for 20 s. 50 µL of this homogenate was then pipetted into 450 µL cold PBS+PIC and placed on ice for ATAC-seq library preparation (see below). The remaining 100 µL homogenate was mixed with 500 µL RLT buffer (Qiagen, Hilden, Germany) with 1% β-mercaptoethanol for use in the Qiagen RNeasy Mini Kit RNA extraction protocol (see below).

## RNAseq library preparation and sequencing

Whole brain RNA was extracted from the 600 µL homogenate in RLT buffer after an additional 30 s homogenization following the Qiagen RNeasy Mini Kit protocol, including a DNase (Qiagen) treatment to remove genomic DNA. RNA quantities were determined for each sample using a Qubit RNA HS Assay Kit (Invitrogen, Carlsbad, CA). High RNA integrity for all samples was confirmed with Bioanalyzer 2100 RNA Pico chips (Agilent, Santa Clara, CA) prior to library preparation.

RNAseq libraries were constructed and sequenced by the W.M. Keck Center for Comparative and Functional Genomics at the Roy J. Carver Biotechnology Center (University of Illinois at Urbana-Champaign). Libraries were constructed from 500 ng RNA per sample using the TruSeq Stranded mRNA HT kit (Illumina, San Diego, CA) on an ePMotion 5075 robot (Eppendorf, Hamburg, Germany). Libraries were uniquely barcoded, quantified, and pooled for sequencing across 6 lanes with 100 nt single-end sequencing on the Illumina HiSeq 4000.

## ATACseq library preparation and sequencing

The 500 µL tissue homogenate was additionally homogenized by aspirating through a 20 gauge needle followed by a 23 gauge needle five times each. Samples were centrifuged at 500 g for 5 min at 4°C. Supernatant was removed, and cells were resuspended in 50 µL cold PBS+PIC. 15 µL of this cell suspension (approximately 1/10$^{th}$ of the total brain,~100k cells) was placed into a new microcentrifuge tube, and this was centrifuged at 500 g for 5 min at 4°C as an additional cell washing step. Supernatant was removed, and cells were gently resuspended in 50 µL cold lysis buffer prepared as in *Buenrostro et al., 2015*. The remainder of the ATACseq library protocol followed *Buenrostro et al., 2015*, with the exception of the final purification step, where a 0.8:1 ratio of Ampure XP beads (Beckman Coulter, Brea, CA) to sample was used to purify each library. In addition to sample libraries, input libraries were constructed from thoracic genomic DNA from a random bee from each colony per sequencing batch using 50 ng of genomic DNA (extracted using the Gentra Puregene Tissue Kit from Qiagen, cat. #158667, following manufacturer's protocol for DNA purification from 25 mg tissue but with 6 µL proteinase K and 4 µL RNase A at the appropriate steps). Genomic DNA was transposed with Nextera Tn5 Transposase (Nextera Kit, Illumina) following the ATACseq protocol immediately following the cell lysis step (*Buenrostro et al., 2015*), again using an 0.8:1 Ampure XP bead clean-up at the end of the protocol. A Qubit dsDNA HS Assay Kit (Invitrogen) was used to quantify each library, and library size and quality was assessed using a Bioanalyzer High-Sensitivity DNA Analysis kit (Agilent).

ATACseq libraries, including input libraries, were pooled at equal nM concentrations and a bead clean-up (0.8:1 ratio of Ampure XP beads to sample) was performed on the pool prior to submission for sequencing. QC on the final pool was performed using qPCR and an AATI Fragment Analyzer by the Keck Center. Libraries were sequenced across three lanes with 100 nt paired-end sequencing on the Illumina HiSeq 4000 by the Keck Center.

## Data processing and analysis

### RNAseq

Sequencing of RNAseq libraries (n = 45, 25 from colony E, 20 from colony F) produced 1,487,641,973 reads which survived quality and adapter trimming using Trimmomatic (version 0.36, parameters used: ILLUMINACLIP: 2:35:30 LEADING:20 TRAILING:20 MINLEN:30). Trimmed reads were aligned to the *A. mellifera* HAv3.1 genome (NCBI accession GCA_003254395.2) using STAR

(version 2.5.3) and default parameters, resulting in an average of 96.7% reads mapping uniquely. The program featureCounts from the Subread package (version 1.5.2) was used to assign mapped reads to gene features from the GFF file from NCBI associated with the *A. mellifera* HAv3.1 genome. On average, 84.8% of uniquely mapped reads were assigned to gene features using featureCounts.

Gene counts were imported into R for differential expression analysis using edgeR. Genes with less than 1 CPM in at least two samples were removed, and remaining count values were normalized using the TMM method. Gene-wise variances were calculated by estimating tagwise dispersions in edgeR on filtered gene count matrices for each group separately and plotted using ggplot2. Tag-wise dispersion estimates were followed by quasi-likelihood F-tests for each pairwise comparison of groups, with FDR correction for multiple testing. Differentially expressed genes (DEGs, FDR < 0.05) for each pairwise comparison are given in *Supplementary file 3*.

## ATACseq

ATACseq libraries (n = 48, 25 from colony E, 20 from colony F, three input libraries) produced 1,110,401,018 paired-end reads which survived quality and adapter trimming using Trimmomatic (version 0.38, parameters used: ILLUMINACLIP: 2:15:10 HEADCROP:10 LEADING:20 TRAILING 20 SLIDINGWINDOW:4:15 MINLEN:30). An average of 98.1% of reads mapped to the *A. mellifera* HAv3.1 genome using bwa mem (version 0.7.17, default parameters). Duplicates were marked and removed prior to further processing using picard (version 2.10.1, average duplication level 30.2%).

Peaks were called from deduplicated BAM files using MACS2 (version 2.1.1, command: callpeak, with parameters: –nomodel -g 2.5e8 –nolambda –keep-dup all –slocal 10000) using the appropriate colony and sequencing batch input as control. Peaks were called on each colony and behavioral group separately, then merged and sorted using BEDTools (version 2.26.0, sort and merge commands). This resulted in a total of 11,614 merged peaks with an average width of 721 bp. Mapped reads were counted to each peak per individual using featureCounts from the Subread package (version 1.5.2). An average of 51.0% of reads were mapped to called peaks.

Peak counts were imported into R for differential accessibility analysis using edgeR. Peaks with less than 1 CPM in at least two samples were removed, and remaining count values were normalized using the TMM method. Gene-wise variances were calculated by estimating tagwise dispersions in edgeR on filtered gene count matrices for each group separately and plotted using ggplot2. Tag-wise dispersion estimates were followed by quasi-likelihood F-tests for each pairwise comparison of groups, with FDR correction for multiple testing. Differentially accessible peak (DAP, FDR < 0.05) results for each pairwise comparison are given in *Supplementary file 4*.

## Functional annotation of differential expression and chromatin accessibility

### Differential expression

Differentially expressed gene (DEG) lists were functionally annotated using Gene Ontology (GO) by first mapping putative orthologs between *A. mellifera* and *Drosophila melanogaster* using reciprocal best BLASTP hits (e-value cutoff = 1e-5). Only DEGs with putative *D. melanogaster* orthologs were included for GO enrichment, and the background list used was all tested genes (those which passed the minimum expression threshold) with putative *D. melanogaster* orthologs. Enrichment tests for biological processes were conducted using GOrilla (*Eden et al., 2009*) with all significant DEGs (FDR < 0.05) against the background list. GO enrichment results for all DEG lists are given in *Supplementary file 3*.

### Differential accessibility

To functionally annotate DAPs, the midpoint coordinates of the 11,614 peaks identified with MACS2 were assigned to genes based on proximity to honey bee gene features (*A. mellifera* HAv3.1 genome). The following features were considered per gene: promoters (1 kb upstream), introns, exons, 5' UTR, 3' UTR, upstream (10 kb upstream), and downstream (10 kb). Peaks not associated with any gene feature were classified as intergenic. When peaks were associated with multiple genes (e.g. the intron of one gene and the promoter of another), they were assigned to individual genes based on the following priority: promoter (highest priority), exon, 5' UTR, 3' UTR, intron, upstream, downstream (lowest priority). If a peak was present in the same highest priority class for multiple

genes, it was randomly assigned to one gene. In this way, each peak was assigned to either a single gene or considered intergenic. Of the 11,614 peaks, 1822 were assigned to the promoter region of a gene, 776 to exons, 1326 to 5' UTRs, 273 to 3' UTRs, 4666 to introns, 1155 to upstream regions, 773 to downstream regions, and 823 peaks were located in intergenic regions.

As before with GO enrichment for DEGs, differentially accessible peaks (DAPs) were functionally annotated by mapping peak-associated genes to putative orthologs in *D. melanogaster* using BLASTP. The background list for enrichment analyses was the list of peaks which met the minimum accessibility count threshold for analysis and which had putative orthologs in *D. melanogaster*. GOrilla (*Eden et al., 2009*) was used for enrichment tests. GO enrichment results for all DAP lists are given in *Supplementary file 4*.

## Motif enrichment of DAPs and DEG regulatory regions

TF motif enrichment analysis in this study was performed similarly to the methods described in *Whitney et al., 2014*. The overall approach is as follows, with details below. For each TF motif, (1) genomic windows were scored for the presence of the motif, (2) window scores were combined into scores for genomic segments of interest, representing either gene regulatory regions or accessibility peaks, (3) a set of motif targets was created using a fixed cutoff on the segment scores, and (4) a statistical test for enrichment was performed between segments that were motif targets and those that were significant in differential analysis.

### Motif scores for genomic windows

First, we divided the honey bee genome (version HAv3.1, NCBI accession GCA_003254395.2) into 500 bp windows with 250 bp shifts. We gathered a collection of 223 representative TFs (*Kapheim et al., 2015*) and downloaded their DNA binding specificities (motifs) characterized as position weight matrices (PWMs) from FlyFactorSurvey (*Zhu et al., 2011*). Separately for each TF motif, we ran the Stubb algorithm (*Sinha et al., 2003*) on all genomic windows to score them for the presence of that TF's binding sites. Tandem repeats in the windows were masked using the Tandem Repeat Finder (*Benson, 1999*) before calculating the Stubb scores to avoid scoring the repeats as weak binding sites. Since the honey bee genome has significant local G/C heterogeneity (*Sinha et al., 2006*), we converted the raw Stubb scores for each window into G/C content-normalized empirical p-values. This was done by determining the rank of each window among all genomic windows of similar G/C content (when grouped into 20 G/C bins).

### Scores for genomic segments

We defined two different collections of genomic segments (accessibility peaks and gene regulatory regions) to analyze with motif enrichment in this study. Since the genomic segments may overlap with a variable number of our genomic windows, we defined a length-adjusted motif score for each segment. This score was calculated using the score of the best scoring window in that segment for the given motif and the number of windows overlapping the segment, as follows: $sc_{seg} = 1 - (1 - pval_{best})^N$ where, $sc_{seg}$ = length-adjusted motif score for the segment, N = number of windows that overlap with the scoring window, and $pval_{best}$ = best G/C normalized empirical p-value among the N overlapping windows.

### Statistical test for TF enrichment

TF enrichment was analyzed for two sets of regions: DAPs (Differentially Accessible Peaks) and DEGs (Differentially Expressed Genes) (*Supplementary file 7*).

For analysis of DAPs, the collection of genomic segments was defined as the combination of all DAPs and randomly selected non-accessible parts of genome that had the same distribution of lengths as those DAPs. The number of randomly selected genomic segments was set to 10 times the number of DAP segments. For each motif, the top 200 scoring segments from the collection were defined as the TF motif target set. Hypergeometric p-values were calculated for each motif-DAP set pair (*Supplementary file 7*) to quantify the significance of the overlap between the corresponding TF motif target set and DAP set.

For DEGs, the collection of genomic segments was the regulatory regions of all genes in the honey bee annotation. Each regulatory region was defined as 5 kb upstream to 2 kb downstream of

the transcriptional start site of its gene (http://veda.cs.uiuc.edu/beeMotifScores/). The top 500 scoring segments from the gene universe were selected as the TF motif target set for each motif. Finally, the significance of the overlap for each motif-DEG set pair (*Supplementary file 7*) was calculated with the Hypergeometric p-value.

All p-values were then converted to q-values using the 'qvalue' function in the R software package qvalue (*Storey et al., 2019*) to control the false discovery rate from multiple hypothesis testing.

For motifs enriched both within DAPs and DEG upstream regions, CentriMo (*Bailey and Machanick, 2012*) from MEME Suite was used to calculate and plot the probability of motif binding across 2 kb windows centered on the peak summit for DAPs and 7 kb windows (5 kb upstream and 2 kb downstream of the transcriptional start site (TSS)) for DEGs. These probabilities are shown in *Figure 3B-C*.

## Individualized gene regulatory network (GRN) analysis

To understand how TFs orchestrate transcriptional changes in the brain, we reconstructed a gene regulatory network (GRN) model using the ASTRIX approach (*Chandrasekaran, 2014*; *Chandrasekaran et al., 2011*). ASTRIX uses gene expression data to identify interactions between TFs and their target genes. The ASTRIX algorithm has been previously used to infer brain GRN models for various organisms including the honey bee (*Bukhari et al., 2017*; *Saul et al., 2017*; *Shpigler et al., 2017*). These models showed significantly high accuracy in predicting gene expression changes in the brain and identified TFs that regulate social behaviors.

Here we applied ASTRIX using the gene expression data of the 45 individual bees along with a list of honey bee TFs as input to identify regulatory interactions. We normalized the transcriptomics data prior to GRN construction using the ComBat algorithm (*Johnson et al., 2007*) to minimize batch and colony effects in the data. The effectiveness of the normalization was checked using PCA. Any TF predicted to interact with a given target gene by ASTRIX had to pass through two criteria: (1) share a significant degree of mutual information with the target gene (p-value$<10^{-6}$), and (2) explain at least 10% of the variance of the target gene, quantified by Least angle regression algorithm. Similarly, each target gene included in the GRN must be predicted with a correlation of at least 0.8 by the ASTRIX model using expression levels of TFs.

The GRN model built by ASTRIX predicted 2190 genes with a Pearson's correlation of 0.8 or higher using expression levels of TFs. Overall, the GRN inferred by ASTRIX contains 4500 interactions between 190 TFs and the 2190 target genes. The full GRN is in *Supplementary file 8*.

To determine TFs correlated with specific behaviors, we first identified genes that were strongly correlated with specific behavior scores across all individuals (FDR p-value of correlation <0.001). TFs whose targets were over-represented among the behavior-correlated genes were then determined. Significance of the overlap between the list of behavior-correlated genes with targets of each TF ('TF module') was estimated using the hypergeometric test.

Finally, to identify TF modules associated with expression changes in each individual ('Individualized TF modules'), genes that were upregulated or downregulated in each individual were identified using z-transformation. Genes in each individual with z-scores above 2 (i.e. two standard deviations above mean) or below −2 were considered to be differentially expressed in an individual. This list of genes was then overlapped with TF modules to identify modules significantly associated with each individual using the hypergeometric test of overlap.

We used a Random Forests classification algorithm for predicting individual behavioral group from TF expression levels. A leave-one-out cross validation analysis was performed wherein the algorithm was trained using data from the remaining 44 individuals and then used to predict the behavior of the 45th individual using its TF levels. The model achieved an accuracy of 82% in predicting behavior. Performance of the model was evaluated by comparison with random shuffling of the behavior labels. We made predictions 100 times with a different set of shuffled labels and compared the accuracy of predictions (i.e. total individuals for which behavioral group was correctly predicted) between the random model and the Random Forest algorithm using a t-test (p=$1\times10^{-8}$). This suggests that TF expression levels can accurately forecast the behavior of the individual, especially for specialists. The relative importance of each TF in predicting behavior was determined using Out-of-bag predictor importance estimation, wherein each predictor's value is permuted and the corresponding impact on model accuracy is determined (importance scores given in *Supplementary file 8*). The random forest classification algorithm was implemented in MATLAB with default parameters

for the number of predictors sampled (square root of the number of predictors, in this case 258 TFs) and default values for the tree depth (n - 1, where n is the training data size).

## Selection of candidate TFs involved in specialized phenotypes

Candidate TFs displayed in *Figure 5* were drawn from multiple analyses presented in this paper and in *Kapheim et al., 2015*. 'Enriched within DAPs' indicates enrichment of the TF motif within forager vs. layer DAPs from the analysis of ATACseq data within this manuscript (see *Motif enrichment of DAPs and DEG promoters* and *Supplementary file 7*). Similarly, 'Enriched near DEGs' indicates enrichment of the TF motif among putative regulatory regions of forager vs. layer DEGs (see *Motif enrichment of DAPs and DEG promoters* and *Supplementary file 7*). 'Module correlated with behavior' indicates that TF module activity is significantly correlated with at least one behavioral metric across individuals (see *Individualized Gene Regulatory Network (GRN) analysis* and *Supplementary file 8*). 'Group Predictive TF' indicates the TF is among the 20 most informative for predicting individual group membership based on TF expression (see *Individualized Gene Regulatory Network (GRN) analysis* and *Supplementary file 8*). 'Implicated in eusocial evolution' indicates that the TF motif was previously found to be associated with social evolution in *Kapheim et al., 2015*.

## Acknowledgements

We thank A Sankey, A Ray, S Bransley, J Cullum, K Wilk and J Falk for assistance in the field, A Hernandez and the staff at the Carver Biotechnology Center for sequencing services, administrators of Biocluster (UIUC) for computational support, and MB Sokolowski, M Hudson, AM Bell, members of the Robinson lab, and three anonymous reviewers for comments that improved this manuscript. Funding: This research was supported by Grant R01GM117467 from the National Institute of General Medical Sciences (GER and N Goldenfeld, PIs), the Christopher Foundation (GER), and the Illinois Sociogenomics Initiative (GER). Data and materials availability: The raw sequence data reported in this paper have been deposited at the National Center for Biotechnology Information (NCBI) Sequence Read Archive, Accession PRJNA593999. Requests for materials should be addressed to BMJ.

## Additional information

### Funding

| Funder | Grant reference number | Author |
| --- | --- | --- |
| National Institute of General Medical Sciences | R01GM117467 | Gene E Robinson |
| Christopher and Dana Reeve Foundation | | Gene E Robinson |
| Illinois Sociogenomics Initiative | | Gene E Robinson |

The funders had no role in study design, data collection and interpretation, or the decision to submit the work for publication.

### Author contributions

Beryl M Jones, Conceptualization, Data curation, Formal analysis, Investigation, Visualization, Methodology, Writing - original draft, Writing - review and editing; Vikyath D Rao, Data curation, Software, Formal analysis, Visualization, Methodology, Writing - review and editing; Tim Gernat, Resources, Data curation, Software, Formal analysis, Investigation, Methodology, Writing - review and editing; Tobias Jagla, Software, Formal analysis, Validation, Methodology; Amy C Cash-Ahmed, Resources, Investigation, Methodology, Writing - review and editing; Benjamin ER Rubin, Software, Methodology, Writing - review and editing; Troy J Comi, Software, Visualization, Methodology, Writing - review and editing; Shounak Bhogale, Software, Investigation, Methodology, Writing - review and editing; Syed S Husain, Software, Formal analysis, Writing - review and editing; Charles Blatti, Software, Formal analysis, Methodology, Writing - review and editing; Martin Middendorf,

Resources, Supervision, Funding acquisition, Writing - review and editing; Saurabh Sinha, Resources, Supervision, Methodology, Writing - review and editing; Sriram Chandrasekaran, Resources, Software, Formal analysis, Supervision, Methodology, Project administration, Writing - review and editing; Gene E Robinson, Conceptualization, Resources, Supervision, Funding acquisition, Writing - original draft, Project administration, Writing - review and editing

### Author ORCIDs
Beryl M Jones ⓘ https://orcid.org/0000-0003-2925-0807
Tim Gernat ⓘ http://orcid.org/0000-0002-5977-3900
Benjamin ER Rubin ⓘ http://orcid.org/0000-0002-6766-0439
Troy J Comi ⓘ https://orcid.org/0000-0002-3215-4026
Martin Middendorf ⓘ http://orcid.org/0000-0002-5426-1092
Sriram Chandrasekaran ⓘ https://orcid.org/0000-0002-8405-5708
Gene E Robinson ⓘ https://orcid.org/0000-0003-4828-4068

### Decision letter and Author response
Decision letter https://doi.org/10.7554/eLife.62850.sa1
Author response https://doi.org/10.7554/eLife.62850.sa2

## Additional files
### Supplementary files
• Supplementary file 1. Daily egg-laying counts, foraging counts, specialist scores, and generalist scores for individual bees.

• Supplementary file 2. Detailed behavioral and physiological information for sequenced bees.

• Supplementary file 3. Differentially expressed genes (DEGs) and Gene Ontology (GO) enrichment of DEGs for each pairwise comparison of specialists and generalists.

• Supplementary file 4. Differentially accessible peaks (DAPs) and Gene Ontology (GO) enrichment of DAPs for each pairwise comparison of specialists and generalists.

• Supplementary file 5. Lists of genes with upper and lower 5% of PC loadings for gene expression and Gene Ontology (GO) enrichment results.

• Supplementary file 6. Lists of genes with upper and lower 5% of PC loadings for chromatin accessibility and Gene Ontology (GO) enrichment results.

• Supplementary file 7. TF motif enrichment of DAPs and DEGs.

• Supplementary file 8. Predicted gene regulatory network (GRN), GRN module correlations with behavior and physiological measurements, and importance scores of TFs from class prediction analysis.

• Supplementary file 9. Overlaps and statistics for comparative gene expression datasets.

• Supplementary file 10. Details of experimental setup for recorded colonies.

• Transparent reporting form

### Data availability
Sequencing data have been deposited at the National Center for Biotechnology Information (NCBI) Sequence Read Archive, Accession PRJNA593999. Additional data generated or analysed during this study are included as supporting files.

The following dataset was generated:

| Author(s) | Year | Dataset title | Dataset URL | Database and Identifier |
|---|---|---|---|---|
| Jones BM, Rao VD, Gernat T, Jagla T, Cash-Ahmed AC, Rubin BE, Comi TJ, | 2020 | RNAseq and ATACseq on honey bee queenless workers (whole brain) | https://www.ncbi.nlm. nih.gov/bioproject/ PRJNA593999/ | NCBI BioProject, PRJNA593999 |

Bhogale S, Husain
SS, Blatti C, Mid-
dendorf M, Sinha S,
Chandrasekaran S,
Robinson GE

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
