## [Decision Letter]

**Acceptance summary:**

This fascinating study assessed gene expression and the role of transcription factors and gene regulation plasticity in honey bees exhibiting predominantly foraging, egg-laying, or generalist behavioral phenotypes. This study uniquely combines sophisticated behavioral analyses with individual gene expression and chromatin profiles, to link together brain and behavior, providing a novel framework for linking gene expression to behavioral plasticity.

**Decision letter after peer review:**

Thank you for submitting your article "Individual differences in honey bee (*Apis mellifera*) behavior enabled by plasticity in brain gene regulatory networks" for consideration by *eLife*. Your article has been reviewed by three peer reviewers, including Lauren A. O’Connell as the Reviewing Editor and Reviewer #1, and the evaluation has been overseen by Christian Rutz as the Senior Editor. The following individual involved in the review of your submission has agreed to reveal their identity: Shannon B. Olsson (Reviewer #3).

The reviewers have discussed their reviews with one another, and the Reviewing Editor has drafted this decision letter to help you prepare a revised submission.

Summary:

This fascinating study assessed gene expression and the role of transcription factors and gene regulatory plasticity in isolated brains of *Apis mellifera* bees exhibiting predominantly foraging, egg-laying, or generalist (both behaviors) behavioral phenotypes. The researchers found that expression of many genes and accessible chromatin differed between the foragers and layers, with generalists sharing profiles from both groups. Through transcription factor motif analysis, the researchers found that motifs were enriched in differential accessibility or in the regulatory regions of genes, and could be used themselves to predict behavior, including correlating with some specific metrics like eggs laid or number of forging trips. Some of these DEGs were also found in species and caste-specific differences shown in previous studies. These results show that behavioral plasticity exists along a continuum of varying molecular pathways; the researchers suggest these results have implications for adaptive colony organization even in queenless colonies, as well as supporting the Ovarian Ground Plan Hypothesis. The data are excellent, but the reviewers have raised a few concerns about interpretations that should be clarified in a revision.

Essential revisions:

1) There was some concern among the reviewers regarding the use of the specialization score. The behaviors were rank scored with -1, 0 and +1 as the maximum points for layer, generalist, and forager, respectively, but it is not clear how this continuum (as shown in Figure 2 and Figure 2—figure supplement 1) was then discretized into the three categories, like in Figure 4—figure supplement 1, for further comparison and to support the statements made in the Discussion. Moreover, the data presented state that gene expression and chromatin availability are correlated with behavior at the individual level (subsection “Brain gene expression and chromatin accessibility are correlated with behavioral variation” and Discussion). However, most of these conclusions are instead based on the specialization score (rather than direct behavioral measurements). While this metric seems great for classifying individuals, using it as a quantitative metric of behavior minimizes the distinctions between different behavioral states and comparisons of behavior within states. For example, in Figure 3E and F, the trends look to be clearly driven by differences between the two specialist groups, but not relationships within behavioral groups. Furthermore, the authors linked genetic variables in TF modules (Supplementary file 8) directly to individual behaviors, which was preferred by some reviewers compared to the specialization score. We encourage the authors to present the corresponding analysis with the RNA-seq and ATAC-seq PCs. We also encourage the authors to consider placing a figure in the main text that relates to Supplementary file 8, as this is an important result (Discussion).

2) One reviewer suggested the terms "plasticity" and "behavioral plasticity" need clarification in the Introduction, and when interpreting the experiments, as sometimes plasticity is confused with genetic variation. There is the potential for genetic differences between these two types of variation, which is both important and very interesting, so the authors should discuss this with more clarity. In particular, the mechanisms underlying behavioral plasticity (e.g. Abstract, Introduction, Discussion), could also be due to genetic differences and this should be clarified. There are certainly ways in which the experiment does reflect unambiguous plasticity, such as the induction of laying behavior after queen removal and the switch between laying and foraging that occurs within generalists. There is scope to discuss some aspects of these data through the lens of plasticity, but the authors need to be much more explicit and focused when they do so.

3) The authors used brain tissue, but several of the genetic correlations they found might not necessarily have to do with neurobiology directly. Indeed, the Introduction starts out with the statement "Behavioral plasticity is of special interest and presents unique challenges, as behavioral traits derive from the integrated actions of genetic, transcriptomic, and neuronal networks". Of course, behavioral traits start with genes, and the brain is needed for responding to many stimuli, but those genes might also influence morphology, biomechanics, physiology, biochemistry, or a host of functions that directly impact behavior. Indeed, the third paragraph of the Introduction mentions the role of gene regulatory networks (GRNs) in the development and evolution of morphological phenotypes. In the Introduction, the authors should consider acknowledging this more explicitly and perhaps even describe some potential gene or GRNs in bees that have been implicated in related behavioral phenotypes, as discussed in the end of the Results. This would set up the narrative better for the Results and Discussion, since several of the gene functions discussed are not necessarily (directly) neuronal in nature.

4) The authors have compared their data in this manuscript to other previously published datasets to make broad claims about the evolution of eusociality. In Figure 6, can the authors explain why the group-predictive TFs seem to not overlap with the list implicated in evolution and what that means for their broad claims in the Discussion?

---

## [Author Response]

Essential revisions:1) There was some concern among the reviewers regarding the use of the specialization score. The behaviors were rank scored with -1, 0 and +1 as the maximum points for layer, generalist, and forager, respectively, but it is not clear how this continuum (as shown in Figure 2 and Figure 2—figure supplement 1) was then discretized into the three categories, like in Figure 4—figure supplement 1, for further comparison and to support the statements made in the Discussion.

Due to variation across days and between colonies, we believe our rank-based approach and resulting specialist/generalist scores is the best way to fairly compare the overall behavioral profiles of individuals. To determine the molecular basis of behavioral phenotypes, however, we made the choice to select from behavioral extremes, including only bees that consistently and frequently engaged in the behaviors of interest. This made discretizing into three categories straightforward (specialist scores near -1 are layers, scores near +1 are foragers, scores near 0 with generalist scores near +1 are generalists), which allowed for group-level, standard statistical approaches to analyze both RNAseq and ATACseq datasets. We have added a clarification about our choice of behavioral extremes and the assignment of these bees to groups:

“Sampled individuals were among those with the most extreme specialist and generalist scores within each colony, and were assigned to behavioral groups based upon their lifetime behavior (Figure 2A).”

Moreover, the data presented state that gene expression and chromatin availability are correlated with behavior at the individual level (subsection “Brain gene expression and chromatin accessibility are correlated with behavioral variation” and Discussion). However, most of these conclusions are instead based on the specialization score (rather than direct behavioral measurements). While this metric seems great for classifying individuals, using it as a quantitative metric of behavior minimizes the distinctions between different behavioral states and comparisons of behavior within states. For example, in Figure 3E and F, the trends look to be clearly driven by differences between the two specialist groups, but not relationships within behavioral groups.

Because we chose extremes for our molecular sampling, we are indeed limited in our ability to make conclusions about within-group variation when using the specialist score as a proxy for behavior. However, the molecular continuum we see for both gene expression and chromatin accessibility (i.e., PCs in what is now Figure 2) is not discretized despite our choice of extremes, and the PCs also show that generalists are molecularly intermediate. We believe this does demonstrate that gene expression and chromatin availability are correlated with behavior overall, but have removed the wording about this being at the individual level in the Abstract and Discussion. These sections now read:

“Brain gene expression and chromatin accessibility profiles were correlated with behavioral variation, with generalists intermediate in behavior and molecular profiles.”

“Beyond these group level differences, we also discovered that large components of this molecular variation were correlated with behavior, and both behavior and brain gene regulatory activity were continuous across bees.”

As these are statements are summaries of all of our results, including the module correlations discussed below which use behavioral measurements themselves (and not the specialist score), we believe they are accurate representations of the data.

Furthermore, the authors linked genetic variables in TF modules (Supplementary file 8) directly to individual behaviors, which was preferred by some reviewers compared to the specialization score. We encourage the authors to present the corresponding analysis with the RNA-seq and ATAC-seq PCs. We also encourage the authors to consider placing a figure in the main text that relates to Supplementary file 8, as this is an important result (Discussion).

The analysis this refers to is the correlation of TF module activity with behavior/physiology, which tested for associations between TF module activity and each of 9 properties of individuals (e.g. eggs laid, foraging trips, proportion of trips with pollen loads). In response to the comments above, we have added a new figure (part of what is now Figure 3) which summarizes a subset of these results (full results are still in Supplementary file 8). We have also changed the order of our Results section to present these analyses immediately following the PCA results. Figures 2 and 3 now illustrate analyses which together speak to the molecular underpinnings of behavioral variation: pairwise differences between groups (Figure 2C-D), correlations between PCs and the individual behavioral aggregate specialist score (Figure 2E-F), and correlations between TF module activity and individual behavioral/physiological metrics (Figure 3A). We believe this new ordering and presentation of results paints a clearer picture of the connections between molecular mechanisms and behavior and thank the reviewers for these very helpful comments.

2) One reviewer suggested the terms "plasticity" and "behavioral plasticity" need clarification in the Introduction, and when interpreting the experiments, as sometimes plasticity is confused with genetic variation. There is the potential for genetic differences between these two types of variation, which is both important and very interesting, so the authors should discuss this with more clarity. In particular, the mechanisms underlying behavioral plasticity (e.g., Abstract, Introduction, Discussion), could also be due to genetic differences and this should be clarified. There are certainly ways in which the experiment does reflect unambiguous plasticity, such as the induction of laying behavior after queen removal and the switch between laying and foraging that occurs within generalists. There is scope to discuss some aspects of these data through the lens of plasticity, but the authors need to be much more explicit and focused when they do so.

We agree that there may be genetic variants influencing the variation in behavior we see across individuals, despite our use of single-drone inseminated (SDI) colonies which reduces genetic variation relative to naturally-mated honey bee colonies. We saw variation in the performance of behaviors both within and across SDI groups, which is suggestive of plasticity not only attributable to genotype, but of course we do not know the contribution of genetic variation to the observed behavioral differences. In the Introduction we mention that subcaste specialization is in part due to genetic differences, and we have added clarification that genetic differences likely also contribute to the behavioral variation we see:

“It is important to note that genetic variation may contribute to individual differences in behavior (Page and Robinson, 1991; Page and Robinson, 1994). However, the induction of egg-laying behavior in queenless colonies is itself a plastic response, suggesting that at least for egg-laying and generalist individuals, a combination of hereditary and environmental factors likely influence the development of these behavioral phenotypes.”

Our primary focus is on how differences in GRN activity (which is influenced by both genetic and non-genetic factors) shape behavioral plasticity. As the reviewer mentions, behavioral plasticity is readily apparent in some individuals (e.g., generalists) while less obvious in others. However, in social insects the colony itself is often discussed as a unit, and laying worker colonies as an aggregate demonstrate a behaviorally plastic response to the loss of the queen. We have carefully reviewed our use of the terms “plasticity” and “behavioral plasticity” throughout and revised the wording to be more clear. In some cases, this was a simple switch from “behavioral plasticity” to “behavioral variation,” since as the reviewer points out not all of our phenotypes are necessarily plastic but could have underlying genetic predispositions. We hope these revisions better reflect the scope of our study. The revised version includes these modified lines:

“We studied the relationship between brain GRN activity and behavior at the individual scale.”

“These results provide new mechanistic insights into the important role played by brain GRNs in the regulation of behavioral variation, with implications for understanding the mechanisms and evolution of complex traits.”

3) The authors used brain tissue, but several of the genetic correlations they found might not necessarily have to do with neurobiology directly. Indeed, the Introduction starts out with the statement "Behavioral plasticity is of special interest and presents unique challenges, as behavioral traits derive from the integrated actions of genetic, transcriptomic, and neuronal networks". Of course, behavioral traits start with genes, and the brain is needed for responding to many stimuli, but those genes might also influence morphology, biomechanics, physiology, biochemistry, or a host of functions that directly impact behavior. Indeed, the third paragraph of the Introduction mentions the role of gene regulatory networks (GRNs) in the development and evolution of morphological phenotypes. In the Introduction, the authors should consider acknowledging this more explicitly and perhaps even describe some potential gene or GRNs in bees that have been implicated in related behavioral phenotypes, as discussed in the end of the Results. This would set up the narrative better for the Results and Discussion, since several of the gene functions discussed are not necessarily (directly) neuronal in nature.

This is an excellent point, especially considering the hormone-related signaling pathways we implicate based on our GRN analyses of behavior. We have added to the Introduction a reference to literature demonstrating how modification of signaling pathways in peripheral tissues can affect brain GRNs and behavior:

“In addition, modification of hormone signaling and GRNs in peripheral tissues has effects on brain GRNs and resulting behavior (Ament et al., 2012).”

We now additionally reference two honey bee papers demonstrating similar tissue cross-talk in the Discussion:

“Given previous demonstration of cross-talk between peripheral tissues and brain gene networks in the honey bee (Ament et al., 2012; Wheeler et al., 2013), our results further suggest that behavioral variation in queenless workers likely involves the coordinated actions of multiple tissue types, including the ovary.”

4) The authors have compared their data in this manuscript to other previously published datasets to make broad claims about the evolution of eusociality. In Figure 6, can the authors explain why the group-predictive TFs seem to not overlap with the list implicated in evolution and what that means for their broad claims in the Discussion?

The candidate TFs shown in Figure 6 (now Figure 5) are derived from multiple analyses and data types, including four in this manuscript and one previously published comparative genomic dataset. We were struck with how much overlap we found in general across these broad data and analysis types, but of course there is not perfect overlap among any given pair of analyses.

The “Group predictive TFs” and “Implicated in eusocial evolution” TFs indeed show little overlap, which is potentially very interesting, but may also be expected given the nature of the analyses. The former is derived from TF activity and its correlation to behavior, while the latter is comparing motif presence across genomes of multiple species of bees. In addition, many TFs (140, see Supplementary file 8) had non-zero importance scores in the predictive analysis, so a reduction to the “most” informative is only part of the complex picture of GRN activity and behavior. Finally, the evolutionary analysis is purely computational, and we do not know whether the activity of these implicated TFs and their targets is also linked to social behavior.

To provide readers with more context and some possible explanations regarding the overlap we do (and do not) see, we have added the following section to the Discussion:

“By combining our analysis of GRNs in individual bees with motif enrichment in gene regulatory regions across individuals, we identified a set of 15 TFs which appear to play a key role in regulating specialist behavioral phenotypes. […] Further research exploring the role of these TFs and their activity in a range of contexts is needed to provide clarity on these results.”